# MASTERING ATARI WITH DISCRETE WORLD MODELS

**Danijar Hafner** *
Google Research

**Timothy Lillicrap**
DeepMind

**Mohammad Norouzi**
Google Research

**Jimmy Ba**
University of Toronto

## ABSTRACT

Intelligent agents need to generalize from past experience to achieve goals in complex environments. World models facilitate such generalization and allow learning behaviors from imagined outcomes to increase sample-efficiency. While learning world models from image inputs has recently become feasible for some tasks, modeling Atari games accurately enough to derive successful behaviors has remained an open challenge for many years. We introduce DreamerV2, a reinforcement learning agent that learns behaviors purely from predictions in the compact latent space of a powerful world model. The world model uses discrete representations and is trained separately from the policy. DreamerV2 constitutes the first agent that achieves human-level performance on the Atari benchmark of 55 tasks by learning behaviors inside a separately trained world model. With the same computational budget and wall-clock time, Dreamer V2 reaches 200M frames and surpasses the final performance of the top single-GPU agents IQN and Rainbow. DreamerV2 is also applicable to tasks with continuous actions, where it learns an accurate world model of a complex humanoid robot and solves stand-up and walking from only pixel inputs.

## 1 INTRODUCTION

To successfully operate in unknown environments, reinforcement learning agents need to learn about their environments over time. World models are an explicit way to represent an agent's knowledge about its environment. Compared to model-free reinforcement learning that learns through trial and error, world models facilitate generalization and can predict the outcomes of potential actions to enable planning (Sutton, 1991). Capturing general aspects of the environment, world models have been shown to be effective for transfer to novel tasks (Byravan et al., 2019), directed exploration (Sekar et al., 2020), and generalization from offline datasets (Yu et al., 2020). When the inputs are high-dimensional images, latent dynamics models predict ahead in an abstract latent space (Watter et al., 2015; Ha and Schmidhuber, 2018; Hafner et al., 2018; Zhang et al., 2019). Predicting compact representations instead of images has been hypothesized to reduce accumulating errors and their small memory footprint enables thousands of parallel predictions on a single GPU (Hafner et al., 2018; 2019). Leveraging this approach, the recent Dreamer agent (Hafner et al., 2019) has solved a wide range of continuous control tasks from image inputs.

Despite their intriguing properties, world models have so far not been accurate enough to compete with the state-of-the-art model-free algorithms on the most competitive benchmarks. The well-established Atari benchmark

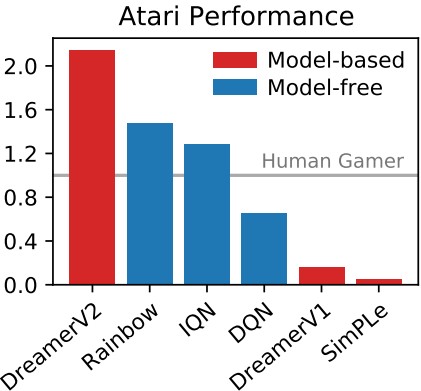

Figure 1: Gamer normalized median score on the Atari benchmark of 55 games with sticky actions at 200M steps. DreamerV2 is the first agent that learns purely within a world model to achieve human-level Atari performance, demonstrating the high accuracy of its learned world model. DreamerV2 further outperforms the top single-GPU agents Rainbow and IQN, whose scores are provided by Dopamine (Castro et al., 2018). According to its authors, SimPLe (Kaiser et al., 2019) was only evaluated on an easier subset of 36 games and trained for fewer steps and additional training does not further increase its performance.

---

*Correspondence to: Danijar Hafner <mail@danijar.com>.

(Bellemare et al., 2013) historically required model-free algorithms to achieve human-level performance, such as DQN (Mnih et al., 2015), A3C (Mnih et al., 2016), or Rainbow (Hessel et al., 2018). Several attempts at learning accurate world models of Atari games have been made, without achieving competitive performance (Oh et al., 2015; Chiappa et al., 2017; Kaiser et al., 2019). On the other hand, the recently proposed MuZero agent (Schrittwieser et al., 2019) shows that planning can achieve impressive performance on board games and deterministic Atari games given extensive engineering effort and a vast computational budget. However, its implementation is not available to the public and it would require over 2 months of computation to train even one agent on a GPU, rendering it impractical for most research groups.

In this paper, we introduce DreamerV2, the first reinforcement learning agent that achieves human-level performance on the Atari benchmark by learning behaviors purely within a separately trained world model, as shown in Figure 1. Learning successful behaviors purely within the world model demonstrates that the world model learns to accurately represent the environment. To achieve this, we apply small modifications to the Dreamer agent (Hafner et al., 2019), such as using discrete latents and balancing terms within the KL loss. Using a single GPU and a single environment instance, DreamerV2 outperforms top single-GPU Atari agents Rainbow (Hessel et al., 2018) and IQN (Dabney et al., 2018), which rest upon years of model-free reinforcement learning research (Van Hasselt et al., 2015; Schaul et al., 2015; Wang et al., 2016; Bellemare et al., 2017; Fortunato et al., 2017). Moreover, aspects of these algorithms are complementary to our world model and could be integrated into the Dreamer framework in the future. To rigorously compare the algorithms, we report scores normalized by both a human gamer (Mnih et al., 2015) and the human world record (Toromanoff et al., 2019) and make a suggestion for reporting scores going forward.

## 2 DREAMERV2

We present DreamerV2, an evolution of the Dreamer agent (Hafner et al., 2019). We refer to the original Dreamer agent as DreamerV1 throughout this paper. This section describes the complete DreamerV2 algorithm, consisting of the three typical components of a model-based agent (Sutton, 1991). We learn the world model from a dataset of past experience, learn an actor and critic from imagined sequences of compact model states, and execute the actor in the environment to grow the experience dataset. In Appendix C, we include a list of changes that we applied to DreamerV1 and which of them we found to increase empirical performance.

### 2.1 WORLD MODEL LEARNING

World models summarize an agent's experience into a predictive model that can be used in place of the environment to learn behaviors. When inputs are high-dimensional images, it is beneficial to learn compact state representations of the inputs to predict ahead in this learned latent space (Watter et al., 2015; Karl et al., 2016; Ha and Schmidhuber, 2018). These models are called latent dynamics models. Predicting ahead in latent space not only facilitates long-term predictions, it also allows to efficiently predict thousands of compact state sequences in parallel in a single batch, without having to generate images. DreamerV2 builds upon the world model that was introduced by PlaNet (Hafner et al., 2018) and used in DreamerV1, by replacing its Gaussian latents with categorical variables.

**Experience dataset** The world model is trained from the agent's growing dataset of past experience that contains sequences of images $x_{1:T}$, actions $a_{1:T}$, rewards $r_{1:T}$, and discount factors $\gamma_{1:T}$. The discount factors equal a fixed hyper parameter $\gamma = 0.999$ for time steps within an episode and are set to zero for terminal time steps. For training, we use batches of $B = 50$ sequences of fixed length $L = 50$ that are sampled randomly within the stored episodes. To observe enough episode ends during training, we sample the start index of each training sequence uniformly within the episode and then clip it to not exceed the episode length minus the training sequence length.

**Model components** The world model consists of an image encoder, a Recurrent State-Space Model (RSSM; Hafner et al., 2018) to learn the dynamics, and predictors for the image, reward, and discount factor. The world model is summarized in Figure 2. The RSSM uses a sequence of deterministic recurrent states $h_t$, from which it computes two distributions over stochastic states at each step. The posterior state $z_t$ incorporates information about the current image $x_t$, while the prior state $\hat{z}_t$ aims to predict the posterior without access to the current image. The concatenation of deterministic and

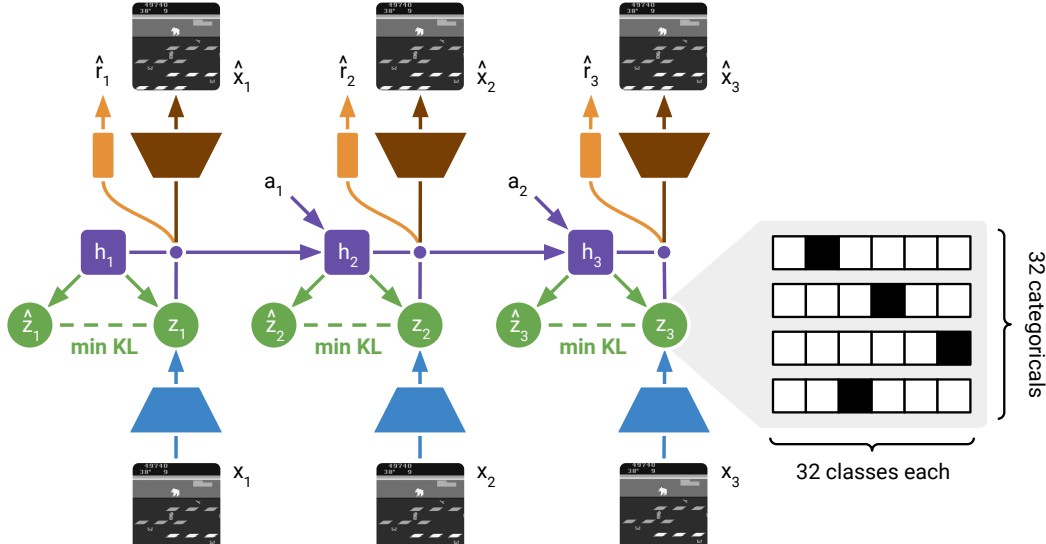

Figure 2: World Model Learning. The training sequence of images $x_t$ is encoded using the CNN. The RSSM uses a sequence of deterministic recurrent states $h_t$. At each step, it computes a posterior stochastic state $z_t$ that incorporates information about the current image $x_t$, as well as a prior stochastic state $\hat{z}_t$ that tries to predict the posterior without access to the current image. Unlike in PlaNet and DreamerV1, the stochastic state of DreamerV2 is a vector of multiple categorical variables. The learned prior is used for imagination, as shown in Figure 3. The KL loss both trains the prior and regularizes how much information the posterior incorporates from the image. The regularization increases robustness to novel inputs. It also encourages reusing existing information from past steps to predict rewards and reconstruct images, thus learning long-term dependencies.

stochastic states forms the compact model state. From the posterior model state, we reconstruct the current image $x_t$ and predict the reward $r_t$ and discount factor $\gamma_t$. The model components are:

$$
\text{RSSM}\begin{cases}
\text{Recurrent model:} & h_t = f_\phi(h_{t-1}, z_{t-1}, a_{t-1}) \\
\text{Representation model:} & z_t \sim q_\phi(z_t \mid h_t, x_t) \\
\text{Transition predictor:} & \hat{z}_t \sim p_\phi(\hat{z}_t \mid h_t) \\
\end{cases}
$$
$$
\begin{aligned}
\text{Image predictor:} & \quad \hat{x}_t \sim p_\phi(\hat{x}_t \mid h_t, z_t) \\
\text{Reward predictor:} & \quad \hat{r}_t \sim p_\phi(\hat{r}_t \mid h_t, z_t) \\
\text{Discount predictor:} & \quad \hat{\gamma}_t \sim p_\phi(\hat{\gamma}_t \mid h_t, z_t).
\end{aligned}
\tag{1}
$$

All components are implemented as neural networks and $\phi$ describes their combined parameter vector. The transition predictor guesses the next model state only from the current model state and the action but without using the next image, so that we can later learn behaviors by predicting sequences of model states without having to observe or generate images. The discount predictor lets us estimate the probability of an episode ending when learning behaviors from model predictions.

**Neural networks**    The representation model is implemented as a Convolutional Neural Network (CNN; LeCun et al., 1989) followed by a Multi-Layer Perceptron (MLP) that receives the image embedding and the deterministic recurrent state. The RSSM uses a Gated Recurrent Unit (GRU; Cho et al., 2014) to compute the deterministic recurrent states. The model state is the concatenation of deterministic GRU state and a sample of the stochastic state. The image predictor is a transposed CNN and the transition, reward, and discount predictors are MLPs. We down-scale the $84 \times 84$ grayscale images to $64 \times 64$ pixels so that we can apply the convolutional architecture of DreamerV1.

---

**Algorithm 1:** Straight-Through Gradients with Automatic Differentiation

```
sample = one_hot(draw(logits))          # sample has no gradient
probs  = softmax(logits)                 # want gradient of this
sample = sample + probs - stop_grad(probs)  # has gradient of probs
```

---

We use the ELU activation function for all components of the model (Clevert et al., 2015). The world model uses a total of 20M trainable parameters.

**Distributions**  The image predictor outputs the mean of a diagonal Gaussian likelihood with unit variance, the reward predictor outputs a univariate Gaussian with unit variance, and the discount predictor outputs a Bernoulli likelihood. In prior work, the latent variable in the model state was a diagonal Gaussian that used reparameterization gradients during backpropagation (Kingma and Welling, 2013; Rezende et al., 2014). In DreamerV2, we instead use a vector of several categorical variables and optimize them using straight-through gradients (Bengio et al., 2013), which are easy to implement using automatic differentiation as shown in Algorithm 1. We discuss possible benefits of categorical over Gaussian latents in the experiments section.

**Loss function**  All components of the world model are optimized jointly. The distributions produced by the image predictor, reward predictor, discount predictor, and transition predictor are trained to maximize the log-likelihood of their corresponding targets. The representation model is trained to produce model states that facilitates these prediction tasks, through the expectation below. Moreover, it is regularized to produce model states with high entropy, such that the model becomes robust to many different model states during training. The loss function for learning the world model is:

$$
\mathcal{L}(\phi) \doteq \mathrm{E}_{q_\phi(z_{1:T} \mid a_{1:T}, x_{1:T})} \Big[ \sum_{t=1}^{T} \underbrace{-\ln p_\phi(x_t \mid h_t, z_t)}_{\text{image log loss}} \underbrace{-\ln p_\phi(r_t \mid h_t, z_t)}_{\text{reward log loss}} \underbrace{-\ln p_\phi(\gamma_t \mid h_t, z_t)}_{\text{discount log loss}}
$$

$$
\underbrace{+\beta \, \mathrm{KL}\big[q_\phi(z_t \mid h_t, x_t) \,\big\|\, p_\phi(z_t \mid h_t)\big]}_{\text{KL loss}} \Big]. \tag{2}
$$

We jointly minimize the loss function with respect to the vector $\phi$ that contains all parameters of the world model using the Adam optimizer (Kingma and Ba, 2014). We scale the KL loss by $\beta = 0.1$ for Atari and by $\beta = 1.0$ for continuous control (Higgins et al., 2016).

**KL balancing**  The world model loss function in Equation 2 is the ELBO or variational free energy of a hidden Markov model that is conditioned on the action sequence. The world model can thus be interpreted as a sequential VAE, where the representation model is the approximate posterior and the transition predictor is the temporal prior. In the ELBO objective, the KL loss serves two purposes: it trains the prior toward the representations, and it regularizes the representations toward the prior. However, learning the transition function is difficult and we want to avoid regularizing the representations toward a poorly trained prior. To solve this problem, we minimize the KL loss faster with respect to the prior than the representations by using different learning rates, $\alpha = 0.8$ for the prior and $1 - \alpha$ for the approximate posterior. We implement this technique as shown in Algorithm 2 and refer to it as KL balancing. KL balancing encourages learning an accurate prior over increasing posterior entropy, so that the prior better approximates the aggregate posterior. KL balancing is different from and orthogonal to beta-VAEs (Higgins et al., 2016).

## 2.2  BEHAVIOR LEARNING

DreamerV2 learns long-horizon behaviors purely within its world model using an actor and a critic. The actor chooses actions for predicting imagined sequences of compact model states. The critic accumulates the future predicted rewards to take into account rewards beyond the planning horizon. Both the actor and critic operate on top of the learned model states and thus benefit from the representations learned by the world model. The world model is fixed during behavior learning, so the actor and value gradients do not affect its representations. Not predicting images during behavior learning lets us efficiently simulate 2500 latent trajectories in parallel on a single GPU.

**Imagination MDP**  To learn behaviors within the latent space of the world model, we define the imagination MPD as follows. The distribution of initial states $\hat{z}_0$ in the imagination MDP is the distribution of compact model states encountered during world model training. From there, the transition predictor $p_\phi(\hat{z}_t \mid \hat{z}_{t-1}, \hat{a}_{t-1})$ outputs sequences $\hat{z}_{1:H}$ of compact model states up to the

---

**Algorithm 2:** KL Balancing with Automatic Differentiation

```
kl_loss =      alpha  * compute_kl(stop_grad(approx_posterior), prior)
        + (1 - alpha) * compute_kl(approx_posterior, stop_grad(prior))
```

---

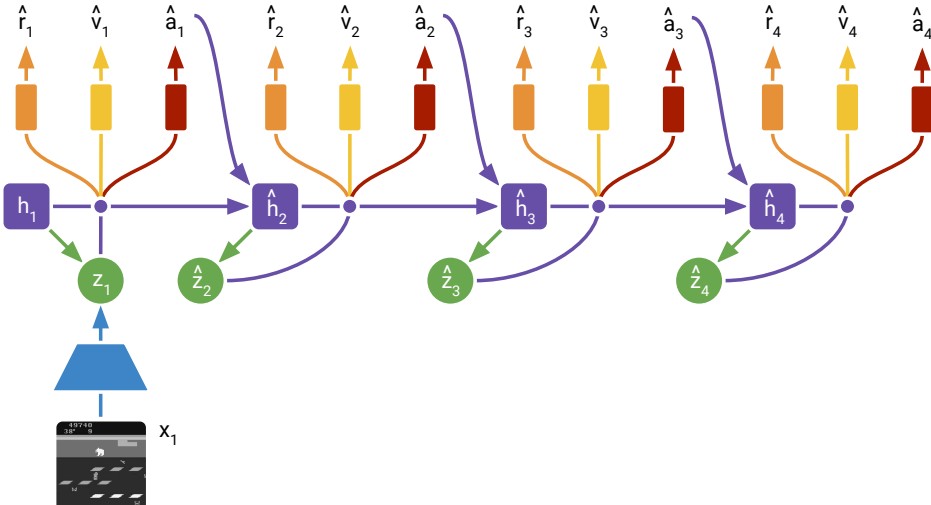

Figure 3: Actor Critic Learning. The world model learned in Figure 2 is used for learning a policy from trajectories imagined in the compact latent space. The trajectories start from posterior states computed during model training and predict forward by sampling actions from the actor network. The critic network predicts the expected sum of future rewards for each state. The critic uses temporal difference learning on the imagined rewards. The actor is trained to maximize the critic prediction, via reinforce gradients, straight-through gradients of the world model, or a combination of them.

imagination horizon $H = 15$. The mean of the reward predictor $p_\phi(\hat{r}_t \mid \hat{z}_t)$ is used as reward sequence $\hat{r}_{1:H}$. The discount predictor $p_\phi(\hat{\gamma}_t \mid \hat{z}_t)$ outputs the discount sequence $\hat{\gamma}_{1:H}$ that is used to down-weight rewards. Moreover, we weigh the loss terms of the actor and critic by the cumulative predicted discount factors to softly account for the possibility of episode ends.

**Model components** To learn long-horizon behaviors in the imagination MDP, we leverage a stochastic actor that chooses actions and a deterministic critic. The actor and critic are trained cooperatively, where the actor aims to output actions that lead to states that maximize the critic output, while the critic aims to accurately estimate the sum of future rewards achieved by the actor from each imagined state. The actor and critic use the parameter vectors $\psi$ and $\xi$, respectively:

$$\begin{aligned} \text{Actor:} \quad & \hat{a}_t \sim p_\psi(\hat{a}_t \mid \hat{z}_t) \\ \text{Critic:} \quad & v_\xi(\hat{z}_t) \approx \mathrm{E}_{p_\phi, p_\psi}\left[\sum_{\tau \geq t} \hat{\gamma}^{\tau - t} \hat{r}_\tau\right]. \end{aligned} \tag{3}$$

In contrast to the actual environment, the latent state sequence is Markovian, so that there is no need for the actor and critic to condition on more than the current model state. The actor and critic are both MLPs with ELU activations (Clevert et al., 2015) and use 1M trainable parameters each. The actor outputs a categorical distribution over actions and the critic has a deterministic output. The two components are trained from the same imagined trajectories but optimize separate loss functions.

**Critic loss function** The critic aims to predict the discounted sum of future rewards that the actor achieves in a given model state, known as the state value. For this, we leverage temporal-difference learning, where the critic is trained toward a value target that is constructed from intermediate rewards and critic outputs for later states. A common choice is the 1-step target that sums the current reward and the critic output for the following state. However, the imagination MDP lets us generate on-policy trajectories of multiple steps, suggesting the use of n-step targets that incorporate reward information into the critic more quickly. We follow DreamerV1 in using the more general $\lambda$-target (Sutton and Barto, 2018; Schulman et al., 2015) that is defined recursively as follows:

$$V_t^\lambda \doteq \hat{r}_t + \hat{\gamma}_t \begin{cases} (1 - \lambda)v_\xi(\hat{z}_{t+1}) + \lambda V_{t+1}^\lambda & \text{if} \quad t < H, \\ v_\xi(\hat{z}_H) & \text{if} \quad t = H. \end{cases} \tag{4}$$

Intuitively, the $\lambda$-target is a weighted average of n-step returns for different horizons, where longer horizons are weighted exponentially less. We set $\lambda = 0.95$ in practice, to focus more on long horizon

targets than on short horizon targets. Given a trajectory of model states, rewards, and discount factors, we train the critic to regress the $\lambda$-return using a squared loss:

$$\mathcal{L}(\xi) \doteq \mathrm{E}_{p_\phi, p_\psi} \left[ \sum_{t=1}^{H-1} \tfrac{1}{2} \big( v_\xi(\hat{z}_t) - \mathrm{sg}(V_t^\lambda) \big)^2 \right]. \tag{5}$$

We optimize the critic loss with respect to the critic parameters $\xi$ using the Adam optimizer. There is no loss term for the last time step because the target equals the critic at that step. We stop the gradients around the targets, denoted by the $\mathrm{sg}(\cdot)$ function, as typical in the literature. We stabilize value learning using a target network (Mnih et al., 2015), namely, we compute the targets using a copy of the critic that is updated every 100 gradient steps.

**Actor loss function**   The actor aims to output actions that maximize the prediction of long-term future rewards made by the critic. To incorporate intermediate rewards more directly, we train the actor to maximize the same $\lambda$-return that was computed for training the critic. There are different gradient estimators for maximizing the targets with respect to the actor parameters. DreamerV2 combines unbiased but high-variance Reinforce gradients with biased but low-variance straight-through gradients. Moreover, we regularize the entropy of the actor to encourage exploration where feasible while allowing the actor to choose precise actions when necessary.

Learning by Reinforce (Williams, 1992) maximizes the actor's probability of its own sampled actions weighted by the values of those actions. The variance of this estimator can be reduced by subtracting the state value as baseline, which does not depend on the current action. Intuitively, subtracting the baseline centers the weights and leads to faster learning. The benefit of Reinforce is that it produced unbiased gradients and the downside is that it can have high variance, even with baseline.

DreamerV1 relied entirely on reparameterization gradients (Kingma and Welling, 2013; Rezende et al., 2014) to train the actor directly by backpropagating value gradients through the sequence of sampled model states and actions. DreamerV2 uses both discrete latents and discrete actions. To backpropagate through the sampled actions and state sequences, we leverage straight-through gradients (Bengio et al., 2013). This results in a biased gradient estimate with low variance. The combined actor loss function is:

$$\mathcal{L}(\psi) \doteq \mathrm{E}_{p_\phi, p_\psi} \left[ \sum_{t=1}^{H-1} \Big( \underbrace{-\rho \ln p_\psi(\hat{a}_t \mid \hat{z}_t)\, \mathrm{sg}(V_t^\lambda - v_\xi(\hat{z}_t))}_{\text{reinforce}} \; \underbrace{-(1-\rho)V_t^\lambda}_{\substack{\text{dynamics} \\ \text{backprop}}} \; \underbrace{-\eta\, \mathrm{H}[a_t|\hat{z}_t]}_{\text{entropy regularizer}} \Big) \right]. \tag{6}$$

We optimize the actor loss with respect to the actor parameters $\psi$ using the Adam optimizer. We consider both Reinforce gradients and straight-through gradients, which backpropagate directly through the learned dynamics. Intuitively, the low-variance but biased dynamics backpropagation could learn faster initially and the unbiased but high-variance could to converge to a better solution. For Atari, we find Reinforce gradients to work substantially better and use $\rho = 1$ and $\eta = 10^{-3}$. For continuous control, we find dynamics backpropagation to work substantially better and use $\rho = 0$ and $\eta = 10^{-4}$. Annealing these hyper parameters can improve performance slightly but to avoid the added complexity we report the scores without annealing.

## 3   EXPERIMENTS

We evaluate DreamerV2 on the well-established Atari benchmark with sticky actions, comparing to four strong model-free algorithms. DreamerV2 outperforms the four model-free algorithms in all scenarios. For an extensive comparison, we report four scores according to four aggregation protocols and give a recommendation for meaningfully aggregating scores across games going forward. We also ablate the importance of discrete representations in the world model. Our implementation of DreamerV2 reaches 200M environment steps in under 10 days, while using only a single NVIDIA V100 GPU and a single environment instance. During the 200M environment steps, DreamerV2 learns its policy from 468B compact states imagined under the model, which is 10,000× more than the 50M inputs received from the real environment after action repeat. Refer to the project website for videos, the source code, and training curves in JSON format.[1]

---

[1] https://danijar.com/dreamerv2

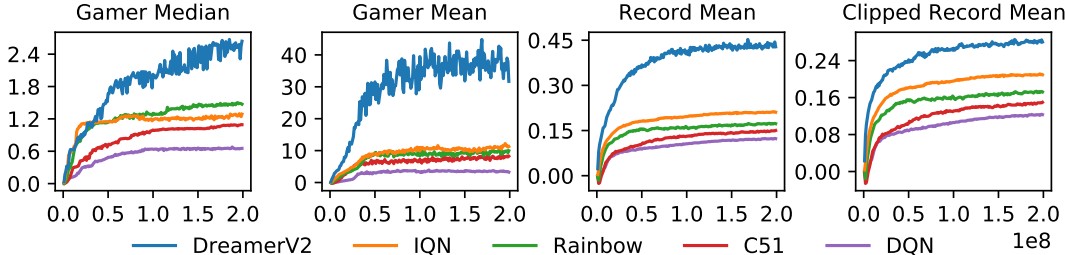

Figure 4: Atari performance over 200M steps. See Table 1 for numeric scores. The standards in the literature to aggregate over tasks are shown in the left two plots. These normalize scores by a professional gamer and compute the median or mean over tasks (Mnih et al., 2015; 2016). In Section 3, we point out limitations of this methodology. As a robust measure of performance, we recommend the metric in the right-most plot. We normalize scores by the human world record (Toromanoff et al., 2019) and then clip them, such that exceeding the record does not further increase the score, before averaging over tasks.

**Experimental setup**  We select the 55 games that prior works in the literature from different research labs tend to agree on (Mnih et al., 2016; Brockman et al., 2016; Hessel et al., 2018; Castro et al., 2018; Badia et al., 2020) and recommend this set of games for evaluation going forward. We follow the evaluation protocol of Machado et al. (2018) with 200M environment steps, action repeat of 4, a time limit of 108,000 steps per episode that correspond to 30 minutes of game play, no access to life information, full action space, and sticky actions. Because the world model integrates information over time, DreamerV2 does not use frame stacking. The experiments use a single-task setup where a separate agent is trained for each game. Moreover, each agent uses only a single environment instance. We compare the algorithms based on both human gamer and human world record normalization (Toromanoff et al., 2019).

**Model-free baselines**  We compare the learning curves and final scores of DreamerV2 to four model-free algorithms, IQN (Dabney et al., 2018), Rainbow (Hessel et al., 2018), C51 (Bellemare et al., 2017), and DQN (Mnih et al., 2015). We use the scores of these agents provided by the Dopamine framework (Castro et al., 2018) that use sticky actions. These may differ from the reported results in the papers that introduce these algorithms in the deterministic Atari setup. The training time of Rainbow was reported at 10 days on a single GPU and using one environment instance.

### 3.1 ATARI PERFORMANCE

The performance curves of DreamerV2 and four standard model-free algorithms are visualized in Figure 4. The final scores at 200M environment steps are shown in Table 1 and the scores on individual games are included in Table K1. There are different approaches for aggregating the scores across the 55 games and we show that this choice can have a substantial impact on the relative performance between algorithms. To extensively compare DreamerV2 to the model-free algorithms, we consider the following four aggregation approaches:

| Agent | Gamer Median | Gamer Mean | Record Mean | Clipped Record Mean |
|---|---|---|---|---|
| DreamerV2 | 2.15 | **42.26** | **0.44** | **0.28** |
| DreamerV2 (schedules) | **2.64** | 31.71 | 0.43 | **0.28** |
| IMPALA | 1.92 | 16.72 | 0.34 | 0.23 |
| IQN | 1.29 | 11.27 | 0.21 | 0.21 |
| Rainbow | 1.47 | 9.95 | 0.17 | 0.17 |
| C51 | 1.09 | 8.25 | 0.15 | 0.15 |
| DQN | 0.65 | 3.28 | 0.12 | 0.12 |

Table 1: Atari performance at 200M steps. The scores of the 55 games are aggregated using the four different protocols described in Section 3. To overcome limitations of the previous metrics, we recommend the task mean of clipped record normalized scores as a robust measure of algorithm performance, shown in the right-most column. DreamerV2 outperforms previous single-GPU agents across all metrics. The baseline scores are taken from Dopamine Baselines (Castro et al., 2018).

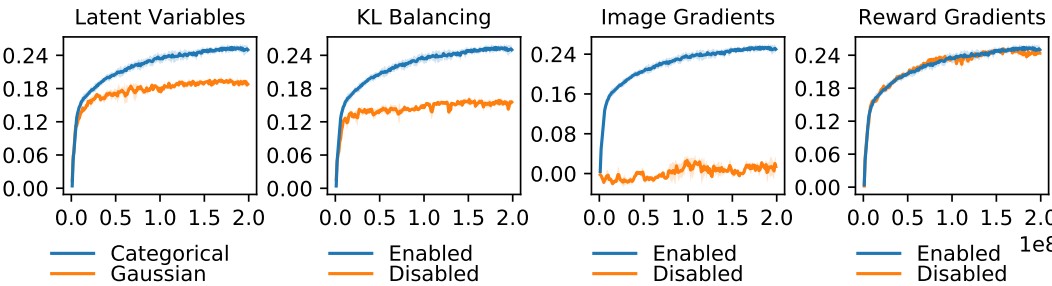

Figure 5: Clipped record normalized scores of various ablations of the DreamerV2 agent. This experiment uses a slightly earlier version of DreamerV2. The score curves for individual tasks are shown in Figure H1. The ablations highlight the benefit of using categorical over Gaussian latent variables and of using KL balancing. Moreover, they show that the world model relies on image gradients for learning its representations. Stopping reward gradients even improves performance on some tasks, suggesting that representations that are not specifically trained to predict previously experienced rewards may generalize better to new situations.

- **Gamer Median** Atari scores are commonly normalized based on a random policy and a professional gamer, averaged over seeds, and the median over tasks is reported (Mnih et al., 2015; 2016). However, if almost half of the scores would be zero, the median would not be affected. Thus, we argue that median scores are not reflective of the robustness of an algorithm and results in wasted computational resources for games that will not affect the score.

- **Gamer Mean** Compared to the task median, the task mean considers all tasks. However, the gamer performed poorly on a small number of games, such as Crazy Climber, James Bond, and Video Pinball. This makes it easy for algorithms to achieve a high normalized score on these few games, which then dominate the task mean so it is not informative of overall performance.

- **Record Mean** Instead of normalizing based on the professional gamer, Toromanoff et al. (2019) suggest to normalize based on the registered human world record of each game. This partially addresses the outlier problem but the mean is still dominated by games where the algorithms easily achieve superhuman performance.

- **Clipped Record Mean** To overcome these limitations, we recommend normalizing by the human world record and then clipping the scores to not exceed a value of 1, so that performance above the record does not further increase the score. The result is a robust measure of algorithm performance on the Atari suite that considers performance across all games.

From Figure 4 and Table 1, we see that the different aggregation approaches let us examine agent performance from different angles. Interestingly, Rainbow clearly outperforms IQN in the first aggregation method but IQN clearly outperforms Rainbow in the remaining setups. DreamerV2 outperforms the model-free agents in all four metrics, with the largest margin in record normalized mean performance. Despite this, we recommend clipped record normalized mean as the most meaningful aggregation method, as it considers all tasks to a similar degree without being dominated by a small number of outlier scores. In Table 1, we also include DreamerV2 with schedules that anneal the actor entropy loss scale and actor gradient mixing over the course of training, which further increases the gamer median score of DreamerV2.

**Individual games** The scores on individual Atari games at 200M environment steps are included in Table K1, alongside the model-free algorithms and the baselines of random play, human gamer, and human world record. We filled in reasonable values for the 2 out of 55 games that have no registered world record. Figure E1 compares the score differences between DreamerV2 and each model-free algorithm for the individual games. DreamerV2 achieves comparable or higher performance on most games except for Video Pinball. We hypothesize that the reconstruction loss of the world model does not encourage learning a meaningful latent representation because the most important object in the game, the ball, occupies only a single pixel. One the other hand, DreamerV2 achieves the strongest improvements over the model-free agents on the games James Bond, Up N Down, and Assault.

| Agent | Gamer Median | Gamer Mean | Record Mean | Clipped Record Mean |
|---|---|---|---|---|
| DreamerV2 | 1.64 | 13.39 | 0.36 | 0.25 |
| No Layer Norm | 1.66 | 11.29 | 0.38 | 0.25 |
| No Reward Gradients | 1.68 | 14.29 | 0.37 | 0.24 |
| No Discrete Latents | 0.85 | 3.96 | 0.24 | 0.19 |
| No KL Balancing | 0.87 | 4.25 | 0.19 | 0.16 |
| No Policy Reinforce | 0.72 | 5.10 | 0.16 | 0.15 |
| No Image Gradients | 0.05 | 0.37 | 0.01 | 0.01 |

Table 2: Ablations to DreamerV2 measured by their Atari performance at 200M frames, sorted by the last column. The this experiment uses a slightly earlier version of DreamerV2 compared to Table 1. Each ablation only removes one part of the DreamerV2 agent. Discrete latent variables and KL balancing substantially contribute to the success of DreamerV2. Moreover, the world model relies on image gradients to learn general representations that lead to successful behaviors, even if the representations are not specifically learned for predicting past rewards.

## 3.2 ABLATION STUDY

To understand which ingredients of DreamerV2 are responsible for its success, we conduct an extensive ablation study. We compare equipping the world model with categorical latents, as in DreamerV2, to Gaussian latents, as in DreamerV1. Moreover, we study the importance of KL balancing. Finally, we investigate the importance of gradients from image reconstruction and reward prediction for learning the model representations, by stopping one of the two gradient signals before entering the model states. The results of the ablation study are summarized in Figure 5 and Table 2. Refer to the appendix for the score curves of the individual tasks.

**Categorical latents**  Categorical latent variables outperform than Gaussian latent variables on 42 tasks, achieve lower performance on 8 tasks, and are tied on 5 tasks. We define a tie as being within $5\%$ of another. While we do not know the reason why the categorical variables are beneficial, we state several hypotheses that can be investigated in future work:

- A categorical prior can perfectly fit the aggregate posterior, because a mixture of categoricals is again a categorical. In contrast, a Gaussian prior cannot match a mixture of Gaussian posteriors, which could make it difficult to predict multi-modal changes between one image and the next.

- The level of sparsity enforced by a vector of categorical latent variables could be beneficial for generalization. Flattening the sample from the 32 categorical with 32 classes each results in a sparse binary vector of length 1024 with 32 active bits.

- Despite common intuition, categorical variables may be easier to optimize than Gaussian variables, possibly because the straight-through gradient estimator ignores a term that would otherwise scale the gradient. This could reduce exploding and vanishing gradients.

- Categorical variables could be a better inductive bias than unimodal continuous latent variables for modeling the non-smooth aspects of Atari games, such as when entering a new room, or when collected items or defeated enemies disappear from the image.

**KL balancing**  KL balancing outperforms the standard KL regularizer on 44 tasks, achieves lower performance on 6 tasks, and is tied on 5 tasks. Learning accurate prior dynamics of the world model is critical because it is used for imagining latent state trajectories using policy optimization. By scaling up the prior cross entropy relative to the posterior entropy, the world model is encouraged to minimize the KL by improving its prior dynamics toward the more informed posteriors, as opposed to reducing the KL by increasing the posterior entropy. KL balancing may also be beneficial for probabilistic models with learned priors beyond world models.

**Model gradients**  Stopping the image gradients increases performance on 3 tasks, decreases performance on 51 tasks, and is tied on 1 task. The world model of DreamerV2 thus heavily relies on the learning signal provided by the high-dimensional images. Stopping the reward gradients increases performance on 15 tasks, decreases performance on 22 tasks, and is tied on 18 tasks. Figure H1 further shows that the difference in scores is small. In contrast to MuZero, DreamerV2 thus learns general representations of the environment state from image information alone. Stopping reward gradients improved performance on a number of tasks, suggesting that the representations that are not specific to previously experienced rewards may generalize better to unseen situations.

| Algorithm | Reward Modeling | Image Modeling | Latent Transitions | Single GPU | Trainable Parameters | Atari Frames | Accelerator Days |
|---|---|---|---|---|---|---|---|
| DreamerV2 | ✓ | ✓ | ✓ | ✓ | 22M | 200M | 10 |
| SimPLe | ✓ | ✓ | ✗ | ✓ | 74M | 4M | 40 |
| MuZero | ✓ | ✗ | ✓ | ✗ | 40M | 20B | 80 |
| MuZero Reanalyze | ✓ | ✗ | ✓ | ✗ | 40M | 200M | 80 |

Table 3: Conceptual comparison of recent RL algorithms that leverage planning with a learned model. DreamerV2 and SimPLe learn complete models of the environment by leveraging the learning signal provided by the image inputs, while MuZero learns its model through value gradients that are specific to an individual task. The Monte-Carlo tree search used by MuZero is effective but adds complexity and is challenging to parallelize. This component is orthogonal to the world model proposed here.

**Policy gradients** Using only Reinforce gradients to optimize the policy increases performance on 18 tasks, decreases performance on 24 tasks, and is tied on 13 tasks. This shows that DreamerV2 relies mostly on Reinforce gradients to learn the policy. However, mixing Reinforce and straight-through gradients yields a substantial improvement on James Bond and Seaquest, leading to a higher gamer normalized task mean score. Using only straight-through gradients to optimize the policy increases performance on 5 tasks, decreases performance on 44 tasks, and is tied on 6 tasks. We conjecture that straight-through gradients alone are not well suited for policy optimization because of their bias.

## 4 RELATED WORK

**Model-free Atari** The majority of agents applied to the Atari benchmark have been trained using model-free algorithms. DQN (Mnih et al., 2015) showed that deep neural network policies can be trained using Q-learning by incorporating experience replay and target networks. Several works have extended DQN to incorporate bias correction as in DDQN (Van Hasselt et al., 2015), prioritized experience replay (Schaul et al., 2015), architectural improvements (Wang et al., 2016), and distributional value learning (Bellemare et al., 2017; Dabney et al., 2017; 2018). Besides value learning, agents based on policy gradients have targeted the Atari benchmark, such as ACER (Schulman et al., 2017a), PPO (Schulman et al., 2017a), ACKTR (Wu et al., 2017), and Reactor (Gruslys et al., 2017). Another line of work has focused on improving performance by distributing data collection, often while increasing the budget of environment steps beyond 200M (Mnih et al., 2016; Schulman et al., 2017b; Horgan et al., 2018; Kapturowski et al., 2018; Badia et al., 2020).

**World models** Several model-based agents focus on proprioceptive inputs (Watter et al., 2015; Gal et al., 2016; Higuera et al., 2018; Henaff et al., 2018; Chua et al., 2018; Wang et al., 2019; Wang and Ba, 2019), model images without using them for planning (Oh et al., 2015; Krishnan et al., 2015; Karl et al., 2016; Chiappa et al., 2017; Babaeizadeh et al., 2017; Gemici et al., 2017; Denton and Fergus, 2018; Buesing et al., 2018; Doerr et al., 2018; Gregor and Besse, 2018), or combine the benefits of model-based and model-free approaches (Kalweit and Boedecker, 2017; Nagabandi et al., 2017; Weber et al., 2017; Kurutach et al., 2018; Buckman et al., 2018; Ha and Schmidhuber, 2018; Wayne et al., 2018; Igl et al., 2018; Srinivas et al., 2018; Lee et al., 2019). Risi and Stanley (2019) optimize discrete latents using evolutionary search. Parmas et al. (2019) combine reinforce and reparameterization gradients. Most world model agents with image inputs have thus far been limited to relatively simple control tasks (Watter et al., 2015; Ebert et al., 2017; Ha and Schmidhuber, 2018; Hafner et al., 2018; Zhang et al., 2019; Hafner et al., 2019). We explain the two model-based approaches that were applied to Atari in detail below.

**SimPLe** The SimPLe agent (Kaiser et al., 2019) learns a video prediction model in pixel-space and uses its predictions to train a PPO agent (Schulman et al., 2017a), as shown in Table 3. The model directly predicts each frame from the previous four frames and receives an additional discrete latent variable as input. The authors evaluate SimPLe on a subset of Atari games for 400k and 2M environment steps, after which they report diminishing returns. Some recent model-free methods have followed the comparison at 400k steps (Srinivas et al., 2020; Kostrikov et al., 2020). However, the highest performance achieved in this data-efficient regime is a gamer normalized median score of 0.28 (Kostrikov et al., 2020) that is far from human-level performance. Instead, we focus on the well-established and competitive evaluation after 200M frames, where many successful model-free algorithms are available for comparison.

**MuZero**    The MuZero agent (Schrittwieser et al., 2019) learns a sequence model of rewards and values (Oh et al., 2017) to solve reinforcement learning tasks via Monte-Carlo Tree Search (MCTS; Coulom, 2006; Silver et al., 2017). The sequence model is trained purely by predicting task-specific information and does not incorporate explicit representation learning using the images, as shown in Table 3. MuZero shows that with significant engineering effort and a vast computational budget, planning can achieve impressive performance on several board games and deterministic Atari games. However, MuZero is not publicly available, and it would require over 2 months to train an Atari agent on one GPU. By comparison, DreamerV2 is a simple algorithm that achieves human-level performance on Atari on a single GPU in 10 days, making it reproducible for many researchers. Moreover, the advanced planning components of MuZero are complementary and could be applied to the accurate world models learned by DreamerV2. DreamerV2 leverages the additional learning signal provided by the input images, analogous to recent successes by semi-supervised image classification (Chen et al., 2020; He et al., 2020; Grill et al., 2020).

## 5    DISCUSSION

We present DreamerV2, a model-based agent that achieves human-level performance on the Atari 200M benchmark by learning behaviors purely from the latent-space predictions of a separately trained world model. Using a single GPU and a single environment instance, DreamerV2 outperforms top model-free single-GPU agents Rainbow and IQN using the same computational budget and training time. To develop DreamerV2, we apply several small modifications to the Dreamer agent (Hafner et al., 2019). We confirm experimentally that learning a categorical latent space and using KL balancing improves the performance of the agent. Moreover, we find the DreamerV2 relies on image information for learning generally useful representations — its performance is not impacted by whether the representations are especially learned for predicting rewards.

DreamerV2 serves as proof of concept, showing that model-based RL can outperform top model-free algorithms on the most competitive RL benchmarks, despite the years of research and engineering effort that modern model-free agents rest upon. Beyond achieving strong performance on individual tasks, world models open avenues for efficient transfer and multi-task learning, sample-efficient learning on physical robots, and global exploration based on uncertainty estimates.

**Acknowledgements**    We thank our anonymous reviewers for their feedback and Nick Rhinehart for an insightful discussion about the potential benefits of categorical latent variables.

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

## A    HUMANOID FROM PIXELS

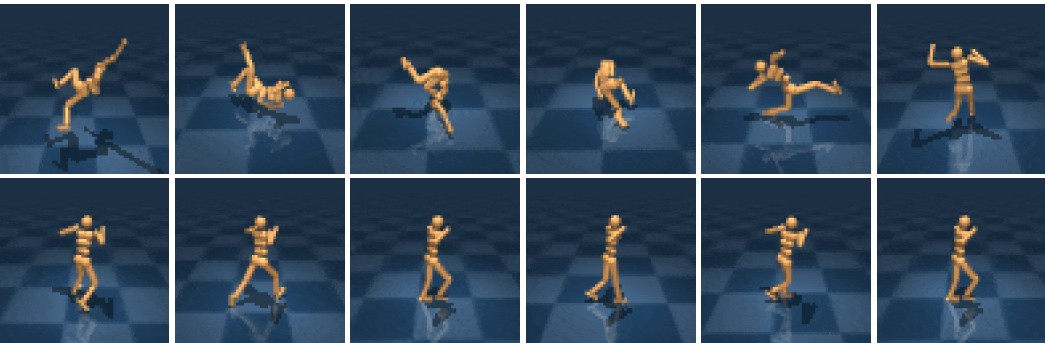

Figure A1: Behavior learned by DreamerV2 on the Humanoid Walk task from pixel inputs only. The task is provided by the DeepMind Control Suite and uses a continuous action space with 21 dimensions. The frames show the agent inputs.

While the main experiments of this paper focus on the Atari benchmark with discrete actions, DreamerV2 is also applicable to control tasks with continuous actions. For this, we the actor outputs a truncated normal distribution instead of a categorical distribution. To demonstrate the abilities of DreamerV2 for continuous control, we choose the challenging humanoid environment with only image inputs, shown in Figure A1. We find that for continuous control tasks, dynamics backpropagation substantially outperforms reinforce gradients and thus set $\rho = 0$. We also set $\eta = 10^{-5}$ and $\beta = 2$ and leave all other hyper parameters at their defaults. We find that DreamerV2 reliably solves both the stand-up motion required at the beginning of the episode and the subsequent walking. The score is shown in Figure A2. To the best of our knowledge, this constitutes the first published result of solving the humanoid environment from only pixel inputs.

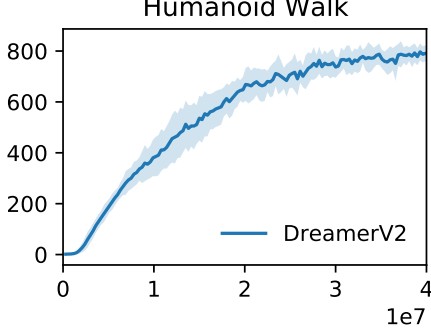

Figure A2: Performance on the humanoid walking task from only pixel inputs.

# B    MONTEZUMA'S REVENGE

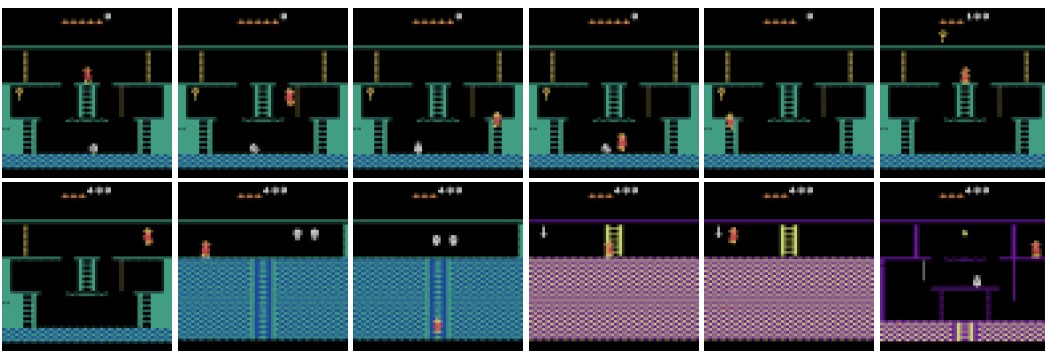

Figure B1: Behavior learned by DreamerV2 on the Atari game Montezuma's Revenge, that poses a hard exploration challenge. Without any explicit exploration mechanism, DreamerV2 reaches about the same performance as the exploration method ICM.

While our main experiments use the same hyper parameters across all tasks, we find that DreamerV2 achieves higher performance on Montezuma's Revenge by using a lower discount factor of $\gamma = 0.99$, possibly to stabilize value learning under sparse rewards. Figure B2 shows the resulting performance, with all other hyper parameters left at their defaults. DreamerV2 outperforms existing model-free approaches on the hard-exploration game Montezuma's Revenge and matches the performance of the explicit exploration algorithm ICM (Pathak et al., 2017) that was applied on top of Rainbow by Taiga et al. (2019). This suggests that the world model may help with solving sparse reward tasks, for example due to improved generalization, efficient policy optimization in the compact latent space enabling more actor critic updates, or because the reward predictor generalizes and thus smooths out the sparse rewards.

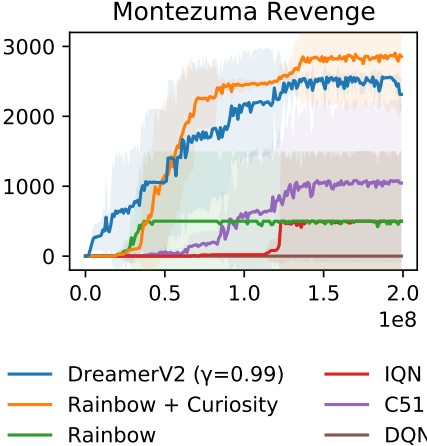

Figure B2: Performance on the Atari game Montezuma's Revenge.

## C   SUMMARY OF MODIFICATIONS

To develop DreamerV2, we used the Dreamer agent (Hafner et al., 2019) as a starting point. This subsection describes the changes that we applied to the agent to achieve high performance on the Atari benchmark, as well as the changes that were tried but not found to increase performance and thus were not not included in DreamerV2.

Summary of changes that were tried and were found to help:

- **Categorical latents**    Using categorical latent states using straight-through gradients in the world model instead of Gaussian latents with reparameterized gradients.
- **KL balancing**    Separately scaling the prior cross entropy and the posterior entropy in the KL loss to encourage learning an accurate temporal prior, instead of using free nats.
- **Reinforce only**    Reinforce gradients worked substantially better for Atari than dynamics backpropagation. For continuous control, dynamics backpropagation worked substantially better.
- **Model size**    Increasing the number of units or feature maps per layer of all model components, resulting in a change from 13M parameters to 22M parameters.
- **Policy entropy**    Regularizing the policy entropy for exploration both in imagination and during data collection, instead of using external action noise during data collection.

Summary of changes that were tried but were found to not help substantially:

- **Binary latents**    Using a larger number of binary latents for the world model instead of categorical latents, which could have encouraged a more disentangled representation, was worse.
- **Long-term entropy**    Including the policy entropy into temporal-difference loss of the value function, so that the actor seeks out states with high action entropy beyond the planning horizon.
- **Mixed actor gradients**    Combining Reinforce and dynamics backpropagation gradients for learning the actor instead of Reinforce provided marginal or no benefits.
- **Scheduling**    Scheduling the learning rates, KL scale, actor entropy loss scale, and actor gradient mixing (from 0.1 to 0) provided marginal or no benefits.
- **Layer norm**    Using layer normalization in the GRU that is used as part of the RSSM latent transition model, instead of no normalization, provided no or marginal benefits.

Due to the large computational requirements, a comprehensive ablation study on this list of all changes is unfortunately infeasible for us. This would require 55 tasks times 5 seeds for 10 days per change to run, resulting in over 60,000 GPU hours per change. However, we include ablations for the most important design choices in the main text of the paper.

## D  HYPER PARAMETERS

| Name | Symbol | Value |
|---|---|---|
| **World Model** | | |
| Dataset size (FIFO) | — | $2 \cdot 10^6$ |
| Batch size | $B$ | 50 |
| Sequence length | $L$ | 50 |
| Discrete latent dimensions | — | 32 |
| Discrete latent classes | — | 32 |
| RSSM number of units | — | 600 |
| KL loss scale | $\beta$ | 0.1 |
| KL balancing | $\alpha$ | 0.8 |
| World model learning rate | — | $2 \cdot 10^{-4}$ |
| Reward transformation | — | $\tanh$ |
| **Behavior** | | |
| Imagination horizon | $H$ | 15 |
| Discount | $\gamma$ | 0.995 |
| $\lambda$-target parameter | $\lambda$ | 0.95 |
| Actor gradient mixing | $\rho$ | 1 |
| Actor entropy loss scale | $\eta$ | $1 \cdot 10^{-3}$ |
| Actor learning rate | — | $4 \cdot 10^{-5}$ |
| Critic learning rate | — | $1 \cdot 10^{-4}$ |
| Slow critic update interval | — | 100 |
| **Common** | | |
| Environment steps per update | — | 4 |
| MPL number of layers | — | 4 |
| MPL number of units | — | 400 |
| Gradient clipping | — | 100 |
| Adam epsilon | $\epsilon$ | $10^{-5}$ |
| Weight decay (decoupled) | — | $10^{-6}$ |

Table D1: Atari hyper parameters of DreamerV2. When tuning the agent for a new task, we recommend searching over the KL loss scale $\beta \in \{0.1, 0.3, 1, 3\}$, actor entropy loss scale $\eta \in \{3 \cdot 10^{-5}, 10^{-4}, 3 \cdot 10^{-4}, 10^{-3}\}$, and the discount factor $\gamma \in \{0.99, 0.999\}$. The training frequency update should be increased when aiming for higher data-efficiency.

# E   AGENT COMPARISON

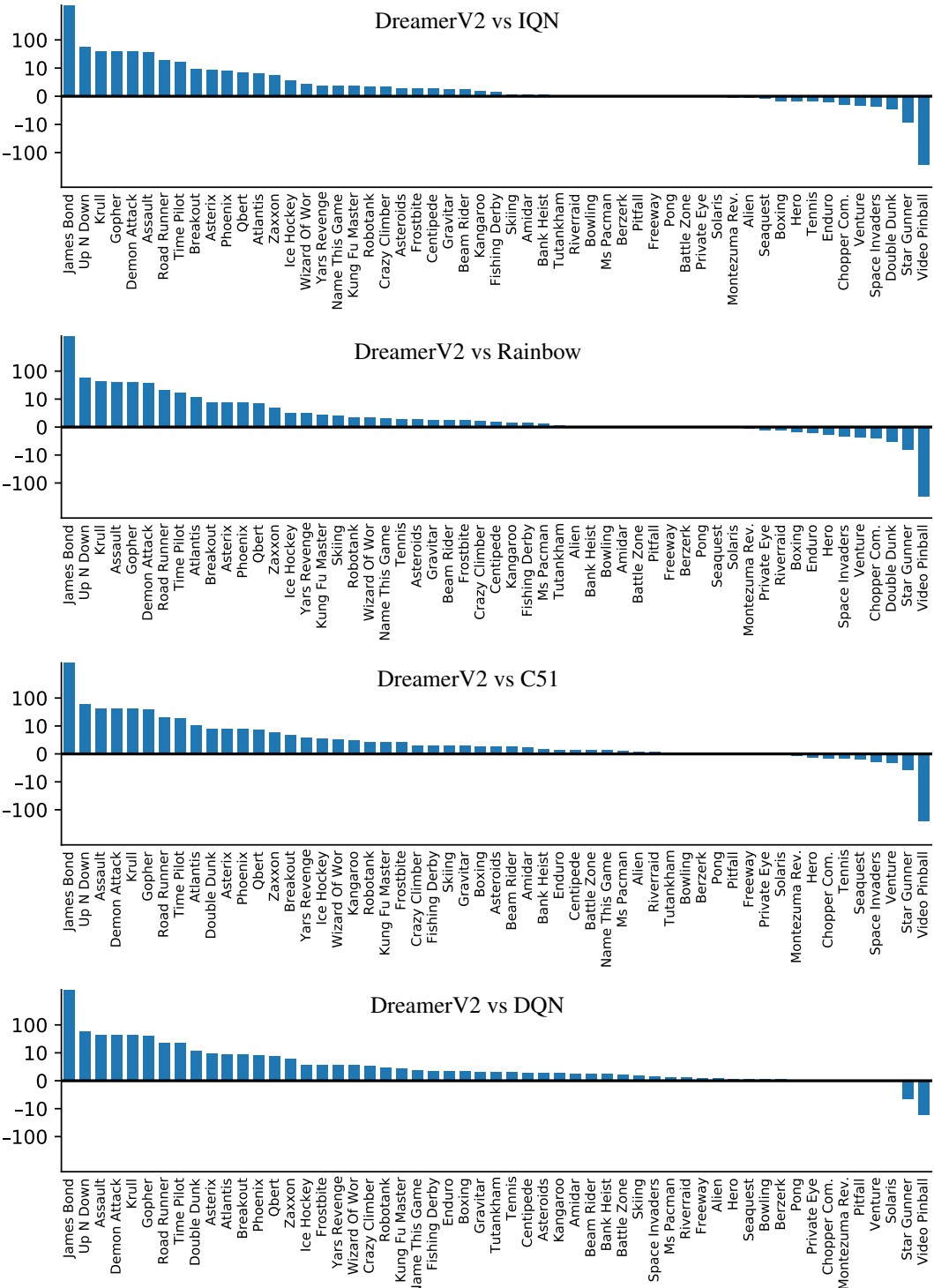

Figure E1: Atari agent comparison. The bars show the difference in gamer normalized scores at 200M steps. DreamerV2 outperforms the four model-free algorithms IQN, Rainbow, C51, and DQN while learning behaviors purely by planning within a separately learned world model. DreamerV2 achieves higher or similar performance on all tasks besides Video Pinball, where we hypothesize that the reconstruction loss does not focus on the ball that makes up only one pixel on the screen.

# F    MODEL-FREE COMPARISON

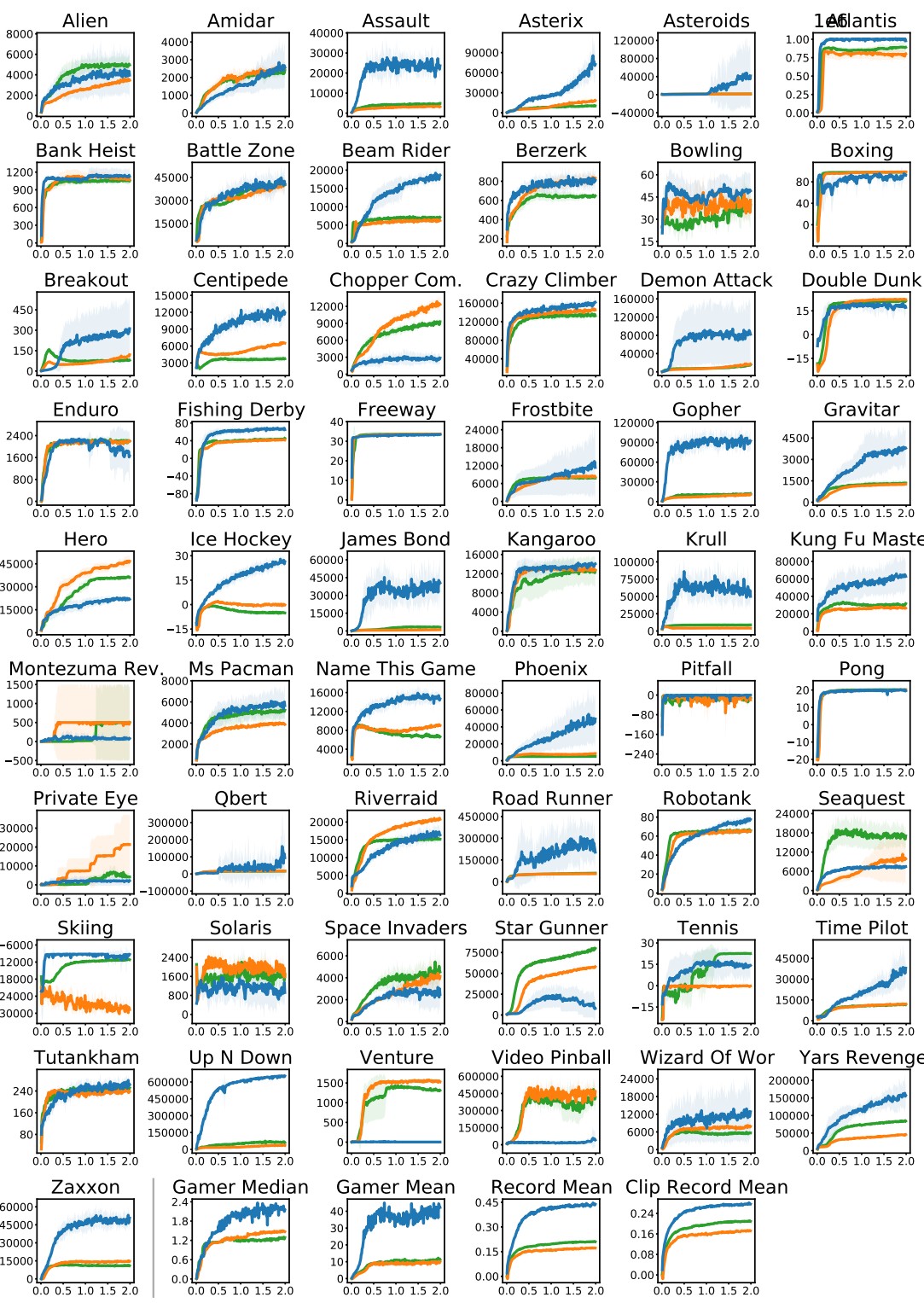

Figure F1: Comparison of DreamerV2 to the top model-free RL methods IQN and Rainbow. The curves show mean and standard deviation over 5 seeds. IQN and Rainbow additionally average each point over 10 evaluation episodes, explaining the smoother curves. DreamerV2 outperforms IQN and Rainbow in all four aggregated scores. While IQN and Rainbow tend to succeed on the same tasks, DreamerV2 shows a different performance profile.

# G  LATENTS AND KL BALANCING ABLATIONS

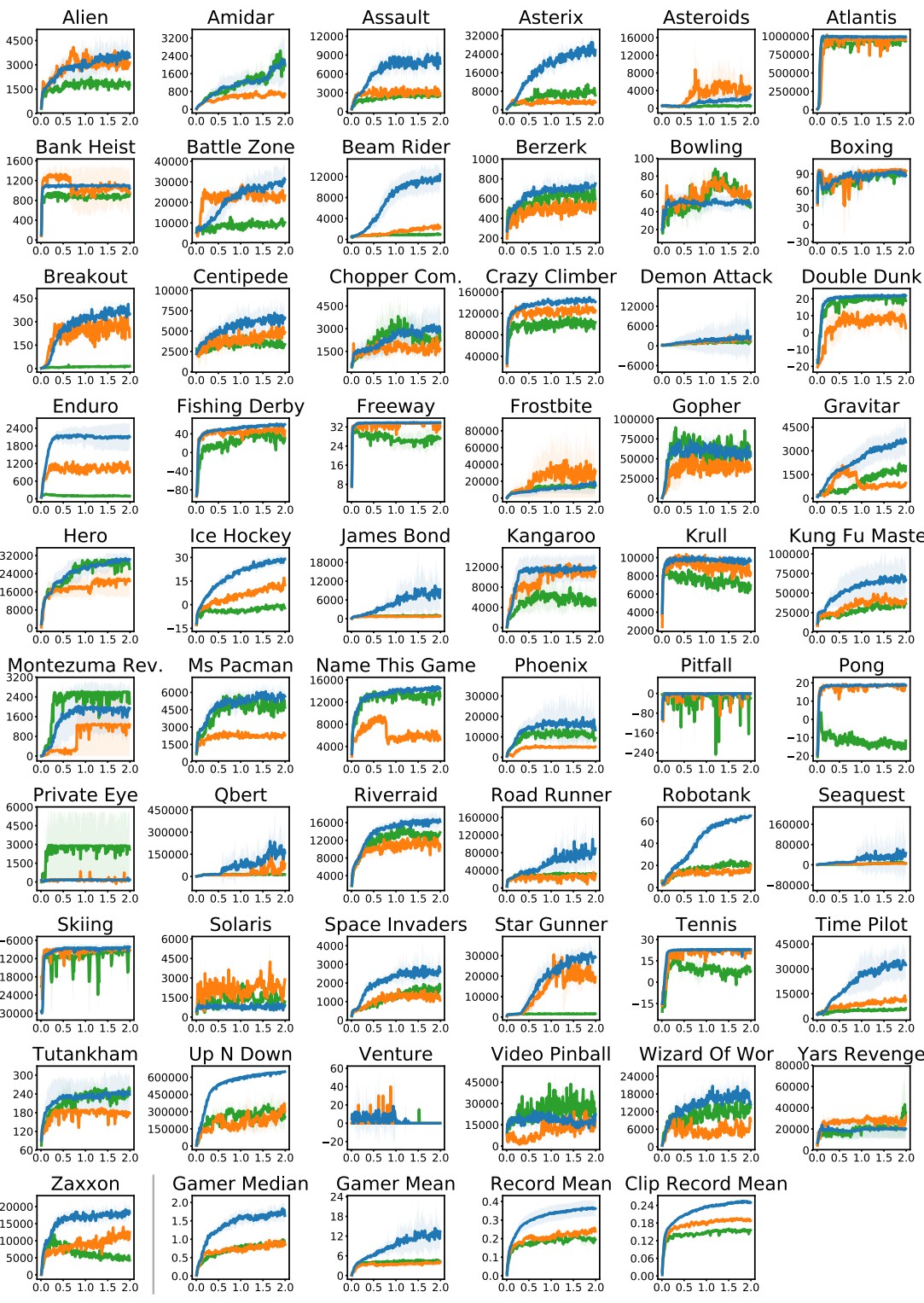

Figure G1: Comparison of DreamerV2, Gaussian instead of categorical latent variables, and no KL balancing. The ablation experiments use a slightly earlier version of the agent. The curves show mean and standard deviation across two seeds. Categorical latent variables and KL balancing both substantially improve performance across many of the tasks. The importance of the two techniques is reflected in all four aggregated scores.

# H    REPRESENTATION LEARNING ABLATIONS

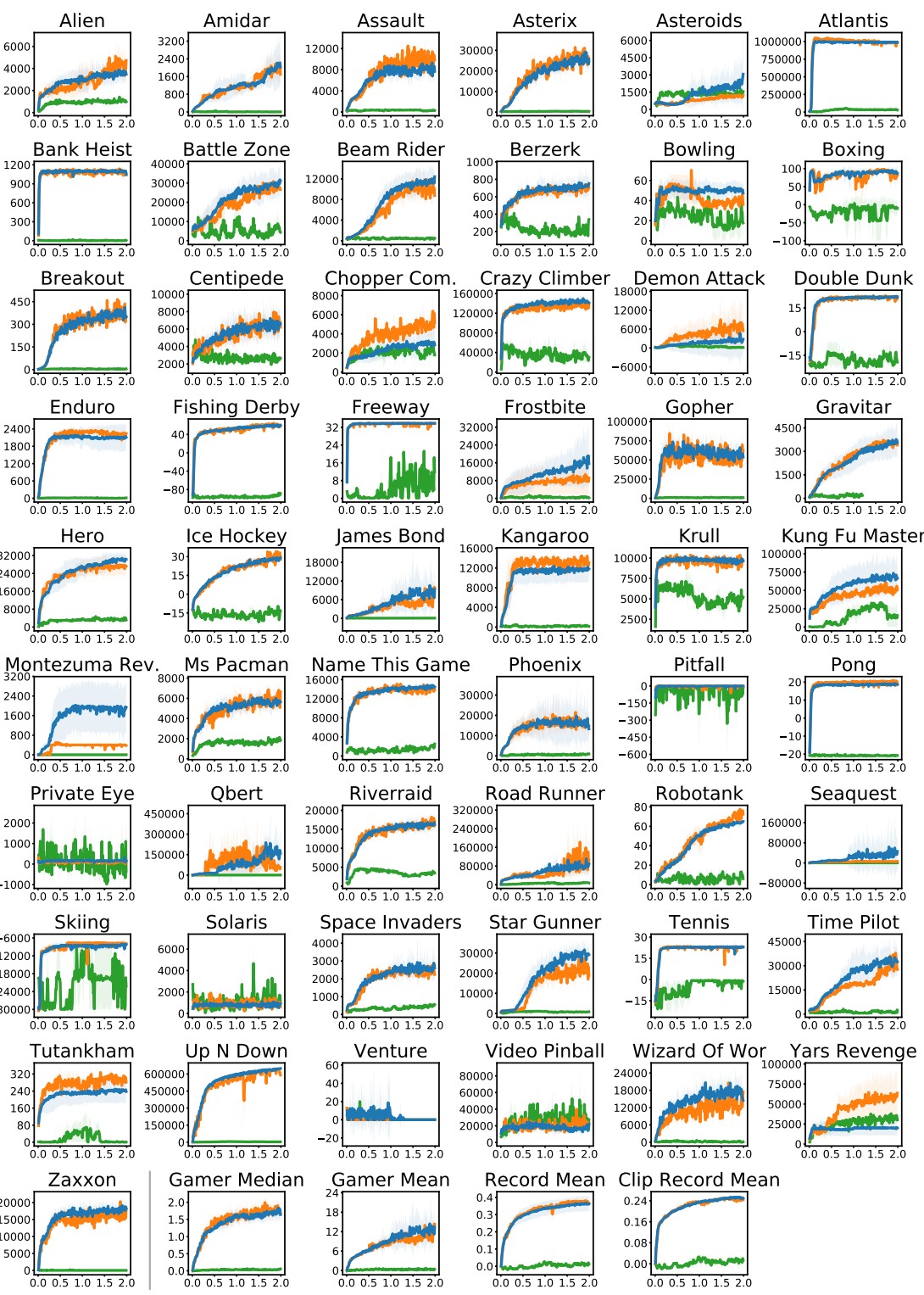

Figure H1: Comparison of leveraging image prediction, reward prediction, or both for learning the model representations. While image gradients are crucial, reward gradients are not necessary for our world model to succeed and their gradients can be stopped. Representations learned purely from images are not biased toward previously encountered rewards and outperform reward-specific representations on a number of tasks, suggesting that they may generalize better to unseen situations.

# I  POLICY LEARNING ABLATIONS

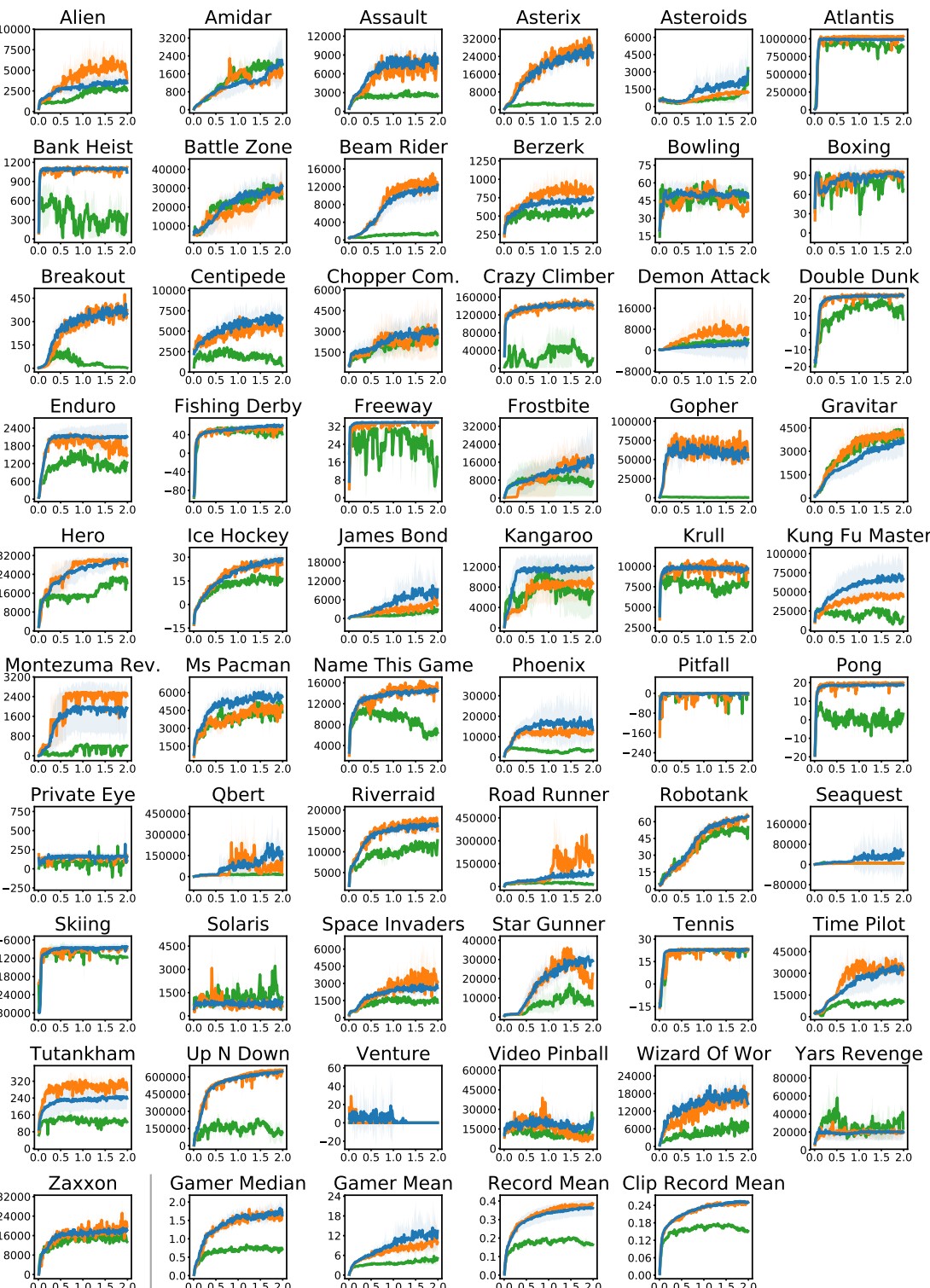

Figure I1: Comparison of leveraging Reinforce gradients, straight-through gradients, or both for training the actor. While Reinforce gradients are crucial, straight-through gradients are not important for most of the tasks. Nonetheless, combining both gradients yields substantial improvements on a small number of games, most notably on Seaquest. We conjecture that straight-through gradients have low variance and thus help the agent start learning, whereas Reinforce gradients are unbiased and help converging to a better solution.

# J ADDITIONAL ABLATIONS

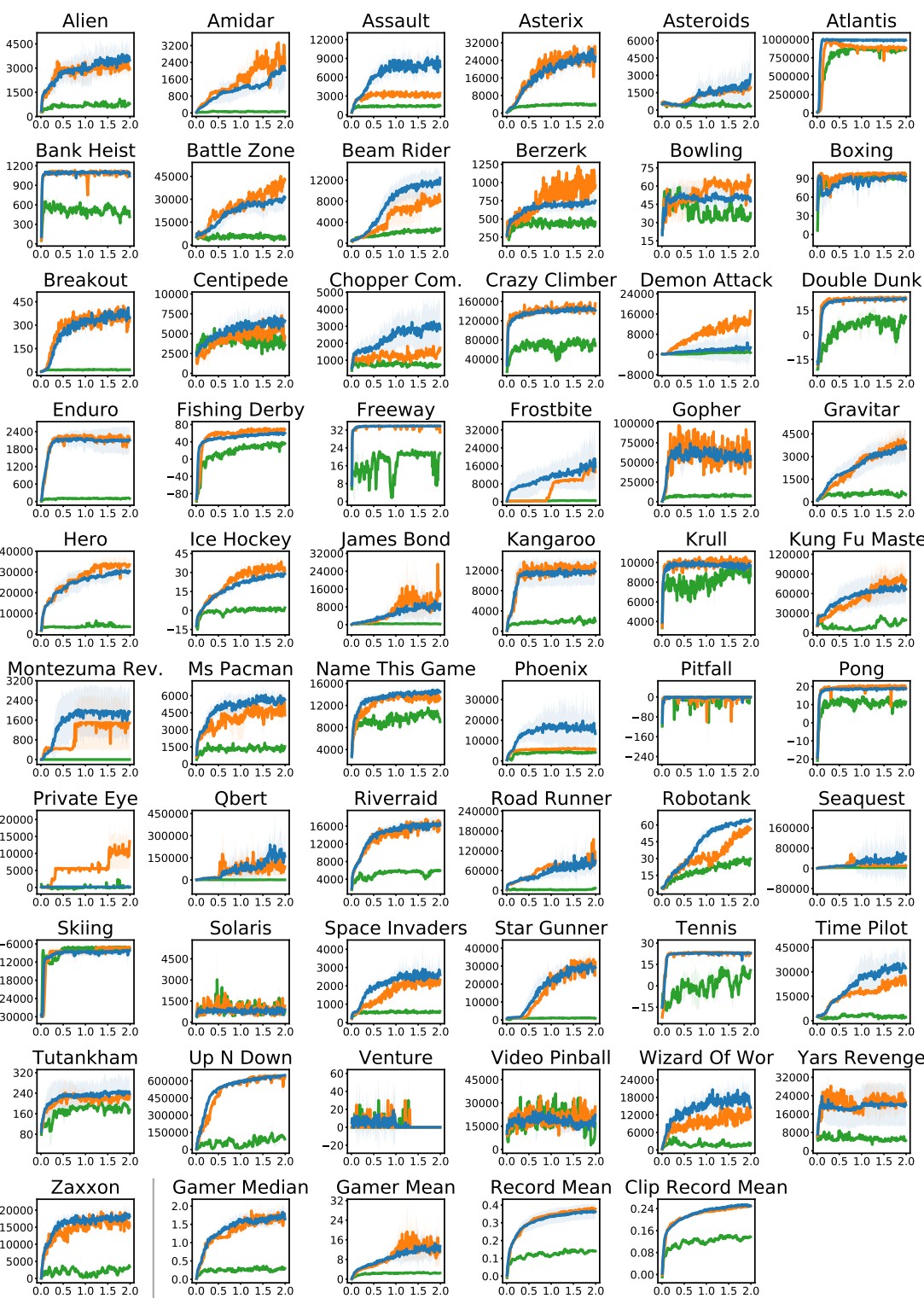

Figure J1: Comparison of DreamerV2 to a version without layer norm in the GRU and to training from experience collected over time by a uniform random policy. We find that the benefit of layer norm depends on the task at hand, increasing and decreasing performance on a roughly equal number of tasks. The comparison to random data collection highlights which of the tasks require non-trivial exploration, which can help guide future work on directed exploration using world models.

## K   ATARI TASK SCORES

| Task | Baselines | | | Algorithms | | |
|------|-----------|------|--------|---------|-----|----------|
| | **Random** | **Gamer** | **Record** | **Rainbow** | **IQN** | **DreamerV2** |
| Alien | 229 | 7128 | 251916 | 3457 | **4961** | 3967 |
| Amidar | 6 | 1720 | 104159 | **2529** | 2393 | **2577** |
| Assault | 222 | 742 | 8647 | 3229 | 4885 | **23625** |
| Asterix | 210 | 8503 | 1000000 | 18367 | 10374 | **72311** |
| Asteroids | 719 | 47389 | 10506650 | 1484 | 1585 | **41526** |
| Atlantis | 12850 | 29028 | 10604840 | 802548 | 890214 | **978778** |
| Bank Heist | 14 | 753 | 82058 | **1075** | 1052 | **1126** |
| Battle Zone | 2360 | 37188 | 801000 | **40061** | **40953** | **40325** |
| Beam Rider | 364 | 16926 | 999999 | 6290 | 7130 | **18646** |
| Berzerk | 124 | 2630 | 1057940 | **833** | 648 | **810** |
| Bowling | 23 | 161 | 300 | 43 | 39 | **49** |
| Boxing | 0 | 12 | 100 | **99** | **98** | 92 |
| Breakout | 2 | 30 | 864 | 120 | 79 | **312** |
| Centipede | 2091 | 12017 | 1301709 | 6510 | 3728 | **11883** |
| Chopper Command | 811 | 7388 | 999999 | **12338** | 9282 | 2861 |
| Crazy Climber | 10780 | 35829 | 219900 | 145389 | 132738 | **161839** |
| Demon Attack | 152 | 1971 | 1556345 | 17071 | 15350 | **82263** |
| Double Dunk | -19 | -16 | 22 | **22** | **21** | 17 |
| Enduro | 0 | 860 | 9500 | **2200** | **2203** | 1656 |
| Fishing Derby | -92 | -39 | 71 | 42 | 45 | **65** |
| Freeway | 0 | 30 | 38 | **34** | **34** | **33** |
| Frostbite | 65 | 4335 | 454830 | 8208 | 7812 | **11384** |
| Gopher | 258 | 2412 | 355040 | 10641 | 12108 | **92282** |
| Gravitar | 173 | 3351 | 162850 | 1272 | 1347 | **3789** |
| Hero | 1027 | 30826 | 1000000 | **46675** | 36058 | 21868 |
| Ice Hockey | -11 | 1 | 36 | 0 | -5 | **26** |
| James Bond | 7 | 29 | 45550 | 1097 | 3166 | **40445** |
| Kangaroo | 52 | 3035 | 1424600 | 12748 | 12602 | **14064** |
| Krull | 1598 | 2666 | 104100 | 4066 | 8844 | **50061** |
| Kung Fu Master | 258 | 22736 | 1000000 | 26475 | 31653 | **62741** |
| Montezuma Revenge | 0 | 4753 | 1219200 | **500** | **500** | 81 |
| Ms Pacman | 307 | 6952 | 290090 | 3861 | 5218 | **5652** |
| Name This Game | 2292 | 8049 | 25220 | 9026 | 6639 | **14649** |
| Phoenix | 761 | 7243 | 4014440 | 8545 | 5102 | **49375** |
| Pitfall | -229 | 6464 | 114000 | -20 | -13 | **0** |
| Pong | -21 | 15 | 21 | **20** | **20** | **20** |
| Private Eye | 25 | 69571 | 101800 | **21334** | 4181 | 2198 |
| Qbert | 164 | 13455 | 2400000 | 17383 | 16730 | **94688** |
| Riverraid | 1338 | 17118 | 1000000 | **20756** | 15183 | 16351 |
| Road Runner | 12 | 7845 | 2038100 | 54662 | 58966 | **203576** |
| Robotank | 2 | 12 | 76 | 66 | 66 | **78** |
| Seaquest | 68 | 42055 | 999999 | 9903 | **17039** | 7480 |
| Skiing | -17098 | -4337 | -3272 | -28708 | -11162 | **-9299** |
| Solaris | 1236 | 12327 | 111420 | 1583 | **1684** | 922 |
| Space Invaders | 148 | 1669 | 621535 | 4131 | **4530** | 2474 |
| Star Gunner | 664 | 10250 | 77400 | 57909 | **80003** | 7800 |
| Tennis | -24 | -8 | 21 | 0 | **23** | 14 |
| Time Pilot | 3568 | 5229 | 65300 | 12051 | 11666 | **37945** |
| Tutankham | 11 | 168 | 5384 | 239 | **251** | **264** |
| Up N Down | 533 | 11693 | 82840 | 34888 | 59944 | **653662** |
| Venture | 0 | 1188 | 38900 | **1529** | 1313 | 2 |
| Video Pinball | 16257 | 17668 | 89218328 | **466895** | 415833 | 41860 |
| Wizard Of Wor | 564 | 4756 | 395300 | 7879 | 5671 | **12851** |
| Yars Revenge | 3093 | 54577 | 15000105 | 45542 | 84144 | **156748** |
| Zaxxon | 32 | 9173 | 83700 | 14603 | 11023 | **50699** |

Table K1: Atari individual scores. We select the 55 games that are common among most papers in the literature. We compare the algorithms DreamerV2, IQN, and Rainbow to the baselines of random actions, DeepMind's human gamer, and the human world record. Algorithm scores are highlighted in bold when they fall within 5% of the best algorithm. Note that these scores are already averaged across seeds, whereas any aggregated scores must be computed before averaging across seeds.

