# OpenReview forum: "Mastering Atari with Discrete World Models"
_ICLR.cc/2021/Conference — ICLR 2021 Poster_

### Official Review · AnonReviewer1 · 2020-10-21
**In my opinion, not enough technical novelty to merit acceptance**

**Rating:** 5
**Confidence:** 4

**Review:**

This paper proposes DreamerV2, a set of modifications to the existing
model-based RL system Dreamer, and shows via an ablation study that
these modifications improve performance on the standard Atari
benchmark over Dreamer. It is shown that DreamerV2 performs well
compared to model-free RL methods as well, especially when using a new
reporting scheme proposed by the authors that normalizes against the
world record on each game rather than the more standard performance of
a strong human player.

In my view, model-based RL is an extremely important and useful
direction to pursue, and so I really appreciate that the authors are
pushing forward in this direction, trying to prove that model-based RL
can be better than model-free RL. However, there are two major reasons
why I lean toward voting reject on this paper. (1) There is simply not
enough technical novelty to warrant publication. (2) The results
are not particularly compelling to me.

(1) It seems that basically, this paper proposes a couple of small
ideas for how to improve Dreamer, the largest one being the use of
categorical, rather than Gaussian, latent states. While it's nice that
these yield improvements (although the categorical latent doesn't help much
by the authors' suggested reporting scheme, looking at the right-most plot in Figure
3), there is ultimately very little in the way of an interesting
technical/scientific contribution in this paper.

(2) Figure 1 greatly surprises me: although it appears on the first
page, the results do not seem commensurately compelling. On the
standard reporting metric shown in the left plot (which I understand
the authors later suggest not to focus on, for various reasons),
DreamerV2 and Rainbow are indistinguishable, and IQN actually
outperforms both in the low-data regime. Looking at the right plot, I
can hardly say that DreamerV2 outperforms Rainbow. On about 50-60% of the games, it outperforms
Rainbow, but on the others, it performs worse. It would be useful to understand *what*
characteristics of the games DreamerV2 is leveraging, on the games
where it outperforms Rainbow. For instance, are these games perhaps
more amenable to model learning for some reason? (The authors do talk
about why Video Pinball doesn't do well, but I'm curious why do some of the
other domains do very well?)

Some other questions:

1. The authors often mention that DreamerV2 uses only a single
environment instance for each game. Some clarification would be useful
surrounding this. How many instances do the other evaluated approaches
use, like Rainbow? Does this only affect training time, or the learning curves
as well?

2. Since you're only focusing on Atari experiments, where we actually
do have a low-dimensional ground-truth model in the form of the ALE
RAM state, I am curious if the authors have tried ablating away the
world model learning phase of their approach, and instead just use
the ground truth ALE transition model? I ask because the authors often
mention that planning in image space is prohibitively expensive, which I totally
agree with, but planning in RAM space (with the known model) may not
be so bad?

---

> ### Author Response · Authors · 2020-11-19
> **Response to AnonReviewer1, Part 2/2**
>
> > It would be useful to understand what characteristics of the games DreamerV2 is leveraging, on the games where it outperforms Rainbow. For instance, are these games perhaps more amenable to model learning for some reason?
>
> We can only speculate about this question. Leveraging learning signal from the images, we conjecture that the world model can help the agent perform better in tasks where the reward signal itself is not very informative, as long as the task does not pose too hard of an exploration challenge. We did not find any correlations between the model loss on the agent's experience dataset and its final human normalized performance.
>
> > The authors often mention that DreamerV2 uses only a single environment instance for each game. Some clarification would be useful surrounding this. How many instances do the other evaluated approaches use, like Rainbow? Does this only affect training time, or the learning curves as well?
>
> Research on the Atari 200M benchmark has historically been performed in two settings, the single environment setting (DQN, PPO, IQN, Rainbow, etc) and the distributed setting (ApeX, R2D2, MuZero, etc). The single environment setting allows one to focus on improvements to the learning algorithm without introducing the complexity of distributed computing and hardware requirements of a distributed algorithm. The algorithms we compare to are also evaluated in the single environment setting.
>
> We leave the development of a distributed DreamerV2 agent for future work and whether it would only affect training time or also the learning dynamics would depend on the details of such an extension.
>
> > Since you're only focusing on Atari experiments, where we actually do have a low-dimensional ground-truth model in the form of the ALE RAM state, I am curious if the authors have tried ablating away the world model learning phase of their approach, and instead just use the ground truth ALE transition model? I ask because the authors often mention that planning in image space is prohibitively expensive, which I totally agree with, but planning in RAM space (with the known model) may not be so bad?
>
> Guo et al. (2014) demonstrated that model-based planning using the true Atari simulator can achieve impressive performance that vastly outperforms the model-free approaches of the time, with the caveat of requiring knowledge of the true dynamics.
>
> We target the well-established Atari 200M benchmark, where the agent only receives pixel inputs and rewards from an unknown environment. This allows for a thorough comparison to strong model-free baselines and makes our algorithm applicable to other environments with unknown dynamics. Our contribution is to develop an accurate and efficient world model that learns directly from high-dimensional inputs, thus enabling successful planning in complex environments with unknown dynamics, as is the case for many real world RL applications.
>
> References:
> - Kaiser et al. Model-Based Reinforcement Learning for Atari. ICLR 2020.
> - Guo et al. Deep learning for real-time Atari game play using offline Monte-Carlo tree search planning. NeurIPS 2014.
> - Hessel et al. Rainbow: Combining Improvements in Deep Reinforcement Learning. 2017.

---

> > ### Comment · AnonReviewer1 · 2020-11-22
> > **response to rebuttal**
> >
> > Thank you for the detailed rebuttal. It has addressed my concerns, and I agree with the authors that certain kinds of empirical research (esp. when small tricks can provide large margins of improvement) could be valuable. That being said, I am generally resistant to the idea that an incremental set of tricks over a prior published approach makes for a good publication in a top-tier venue like ICLR. I recognize that this may be unfair to the authors, who have clearly put a lot of effort into finding the *right* tricks to employ (Appendix A is good evidence of this). Also, there are other contributions in the paper such as the suggestion of the new reporting scheme. So, let's go with the following compromise. I will update my score to a 5 (leaning reject), but during the discussion phase with other reviewers, I will be happy to go along with an accept recommendation if others feel that there is sufficient interest & reason to publish this work.

---

> > > ### Author Response · Authors · 2020-11-23
> > > **Follow-up response to AnonReviewer1**
> > >
> > > We thank you for your response, for revising your evaluation of our paper, and more generally for participating in the discussion!
> > >
> > > As you pointed out as well, the main contribution of our paper is to substantially improve the empirical performance of world models for reinforcement learning. Namely, DreamerV2 achieves 11 times higher asymptotic performance than SimPLE on the Atari benchmark. Moreover, Table 2 shows that removing either discrete latents or KL balancing results in a performance drop by 46%. This demonstrates a strong improvement over DreamerV1, which used neither of the two techniques, even without considering the other changes documented in Appendix A.
> > >
> > > We believe that this empirical contribution, in addition to our technical contributions of KL balancing and a world model that leverages a vector of multiple categorical latent variables, and the proposed reporting scheme, provide substantial value to the research community that can accelerate future research on world models.

---

> ### Author Response · Authors · 2020-11-19
> **Response to AnonReviewer1, Part 1/2**
>
> Thank you for your review! Our paper focuses on developing an accurate and efficient world model for Atari, a goal we achieve with a 11x improvement in final performance over SimPLE (Kaiser et al., 2019). We also show a clear improvement over Rainbow in all metrics now, after having updated two hyper parameters and included more random seeds. Finally, we have added comprehensive ablations to understand the contributions of the individual components of the world model. We discuss these points in detail below. Please let us know if this fully addresses your concerns, or if there are any other issues that we should address.
>
> > In my view, model-based RL is an extremely important and useful direction to pursue, and so I really appreciate that the authors are pushing forward in this direction, trying to prove that model-based RL can be better than model-free RL.
>
> We agree!
>
> > It seems that basically, this paper proposes a couple of small ideas for how to improve Dreamer, the largest one being the use of categorical, rather than Gaussian, latent states. While it's nice that these yield improvements [...] there is ultimately very little in the way of an interesting technical/scientific contribution in this paper.
>
> We completely agree that our paper is of mostly empirical nature. We also believe that empirical papers that demonstrate the surprisingly high performance of a simple algorithm can be impactful on the research community. Additionally, we highlight the use of discrete latent variables in a world model that are trained by straight-through gradients and the idea of KL balancing as technical and actionable contributions.
>
> To further support these technical contributions, we have added an ablation study that is summarized in the experiments section. The results confirm the empirical importance of discrete latents and KL balancing. Additionally, we ablated using only image or reward gradients for learning the model representations rather than both. We find that DreamerV2 heavily relies on image gradients. Interestingly, removing reward gradients improved performance across several games, suggesting that representations that are less specific to previously received rewards may generalize better.
>
> > Figure 1 greatly surprises me: although it appears on the first page, the results do not seem commensurately compelling. On the standard reporting metric shown in the left plot [...] DreamerV2 and Rainbow are indistinguishable [...]. Looking at the right plot, I can hardly say that DreamerV2 outperforms Rainbow. On about 50-60% of the games, it outperforms Rainbow, but on the others, it performs worse.
>
> We have run additional experiments and DreamerV2 now clearly outperforms Rainbow in the plot of Figure 1 as well as in Table 1. The gamer normalized median scores are as follows: DreamerV2 (1.59), Rainbow (1.47), and IQN (1.32). The change we made for this was to update two hyper parameters. Furthermore, our paper focuses on developing an accurate and efficient world model that performs well when combined with a simple policy optimization scheme, to which the more sophisticated ideas for policy optimization of Rainbow and IQN could be added.
>
> Besides this, we would like to make a broader point. Modern model-free RL approaches, such as Rainbow and IQN, rest upon years of research and engineering effort to achieve their high performance. World models, in contrast, have only recently started becoming competitive in reinforcement learning from high-dimensional inputs. Even matching the performance of top model-free approaches using world models has been an important goal of the community that has had several failed attempts, as described in our introduction. Nonetheless, world models are an important research direction due to their potential for transfer to unseen tasks, directed exploration, and generalization.
>
> As such, the goal of our paper is to advance world models for reinforcement learning. We achieve this goal by substantially outperforming the asymptotic performance of SimPLE (Kaiser et al., 2019) with a gamer normalized median score of 1.59 compared to 0.14. Kaiser et al. describe in their Section 6.2 that SimPLE's performance only increases up to 500k agent steps (2M environment steps). As such, our paper is the first to demonstrate human-level Atari performance through a separately trained world model and improves over the final performance of SimPLE by a factor of 11x. We believe that this result alone offers substantial benefit to the research community.
>
> > although the categorical latent doesn't help much by the authors' suggested reporting scheme, looking at the right-most plot in Figure 3
>
> Thank you for pointing this out. We have updated the ablations with experiments for 100M steps rather than the 45M steps in our initial submission. The difference between categorical and Gaussian latent variables is very clear now.

---

### Official Review · AnonReviewer3 · 2020-10-28
**Impressive empirical results. More analysis would make it better.**

**Rating:** 8
**Confidence:** 5

**Review:**

Summary:
The focus of this paper is to extend DreamerV1 to perform harder control tasks such as Atari instead of easier control tasks which were demonstrated in DreamerV1. In doing so, this paper proposes a model-based RL approach which is
- the first Atari agent that achieves human-level performance on the Atari benchmark of 55 tasks by learning behaviors inside a separately trained world model. Human-level performance historically required model-free agents.
- appears to be the first Atari agent which uses a stochastic recurrent state to model the observations and train the policy purely via latent imagination. Consequently, it is much more efficient in terms of training computation in comparison to previous model-based RL for Atari approaches (e.g. SimPLE).
- outperforms model-free single-GPU Atari approaches such as Rainbow and IQN which rest on years of research. This suggests great promise for model-based approaches.

Novelty of the problem:
- Cons: The claims of the paper mainly apply in the context of Atari + single-GPU setup.
- Pros: That being said, it is an important setup from the perspective of model-based RL.

Contributions in terms of analysis:
- Cons: There could be more analysis about what aspects of Atari games make it different from DreamerV1 tasks, and in this context, why Categorical latents provide a better world model. What are the current limitations of DreamerV2 in terms of Atari? What lessons can we learn about applying it outside Atari / single-GPU setting.

Contribution of the experiments:
- Pros: Experiments provide valuable proof of concept, showing that model-based RL can outperform top model-free algorithms on Atari benchmark despite years of research and engineering effort. 2) DreamerV2 is also computationally efficient. 3) Provides a good analysis of which metric is appropriate for comparing performance i.e. Mean Clipped Record Normalized.

Novelty of the conceptual idea for the solution:
- Pros: The model is elegant. For Atari, it appears to be the first model-based approach that uses a recurrent state space model (RSSM) for modeling observations and reward in a stochastic framework. This makes it computationally much faster than previous model-based approaches. The closed-form straight-through gradients for training the policy is also a nice characteristic.
- Cons: Much of this is already proposed in DreamerV1.

Novelty of the low-level implementation of the conceptual idea:
- Pros: The use of Discrete latents versus Gaussian is interesting. The argument for KL balancing appears convincing.
- Cons: However other innovations are small and not analyzed in detail.

Presentation:
- Pros: Well-written
- Suggestions: Pseudo-code of the training loop may be helpful. Qualitative examples of imagination may be helpful.

---- After Rebuttal ----
After reading other reviews and the rebuttal, I stay at my current score. Given that the paper does not contain much analysis about "why", the paper is about an empirical discovery of a mode that is critical in getting good performance. I think this kind of discovery paper is also important to share with the community.

---

> ### Author Response · Authors · 2020-11-19
> **Response to AnonReviewer3, Part 2/2**
>
> > What lessons can we learn about applying it [...] single-GPU setting
>
> DreamerV2 is amenable to a distributed implementation. The three components of its training loop are data collection, world model training, and policy training. The components could be run in parallel to reduce wall clock time by leveraging multiple environment instances and machines, following the example of existing distributed Atari agents (e.g. Kapturowski et al., 2019).
>
> > Qualitative examples of imagination may be helpful.
>
> We have made available multi-step video predictions for hold out sequences at this anonymous link: https://imgur.com/a/fgCkzPu. For these, the world model receives the first 5 frames and then predicts forward open-loop for 45 frames given the action sequence. The top row shows 6 hold out sequences, the middle row the corresponding model predictions, and the bottom row shows pixel difference between the two. For the final version of the paper, we will also train a DreamerV2 world model on RGB images of the Atari environments to generate more visually appealing video predictions. Our reported scores all follow the standard Atari evaluation protocol that prescribes grayscale images.
>
> > Suggestions: Pseudo-code of the training loop may be helpful.
>
> That's a great idea. We will add it to the paper.
>
> References:
> - Kapturowski et al. Recurrent Experience Replay in Distributed Reinforcement Learning. ICLR 2019.

---

> ### Author Response · Authors · 2020-11-19
> **Response to AnonReviewer3, Part 1/2**
>
> Thank you for your review and the actionable suggestions. To study the effects of the individual ingredients of our world model, we have conducted an ablation study that is summarized in the experiments section, with full training curves in the appendix. We respond to your questions below in detail. Please let us know if this fully addresses your concerns, or if there are other issues that we should address.
>
> > Summary: The focus of this paper is to extend DreamerV1 to perform harder control tasks such as Atari instead of easier control tasks which were demonstrated in DreamerV1. In doing so, this paper proposes a model-based RL approach which is
> the first Atari agent that achieves human-level performance on the Atari benchmark of 55 tasks by learning behaviors inside a separately trained world model. Human-level performance historically required model-free agents.
> appears to be the first Atari agent which uses a stochastic recurrent state to model the observations and train the policy purely via latent imagination. Consequently, it is much more efficient in terms of training computation in comparison to previous model-based RL for Atari approaches (e.g. SimPLE).
> outperforms model-free single-GPU Atari approaches such as Rainbow and IQN which rest on years of research. This suggests great promise for model-based approaches.
>
> Thank you for this accurate summary of our paper!
>
> > The main weakness: I wish we gained a better understanding of why the discrete latent space works well. Interpretability might be difficult (and I don't necessarily consider that bad -- the results speak for themselves) but we are left with a mix of several plausible hypotheses. Perhaps some visualization might be useful.
>
> We completely agree that understanding in which environments the components of our algorithm, and especially the choice of discrete latent variables, are beneficial is interesting. We have updated the paper to include the learning curves for discrete and continuous latents for the 55 individual tasks. We find that discrete latents match or outperform continuous latents on almost all games (the exceptions being Asteroids, Demon Attack, Frostbite, Solaris). This result suggests that the benefit of discrete over continuous latents is not very specific to properties of individual games within the Atari benchmark.
>
> An interesting direction to further investigate the effects of discrete latent variables would be to leverage the offline experience datasets of Agarwal et al. (2020) or Gulcehre et al. (2020) to compare the accuracy of model predictions. This would enable separating the effects of model accuracy from those of improved policy optimization and exploration. We leave such a study for future work.
>
> > Novelty of the low-level implementation of the conceptual idea: Pros: The use of Discrete latents versus Gaussian is interesting. The argument for KL balancing appears convincing. Cons: However other innovations are small and not analyzed in detail.
>
> To investigate this question, we have conducted an ablation study that evaluates 4 key ingredients of our world model. In summary, we identify discrete latents, KL balancing, and image reconstruction as crucial components of the algorithm. Interestingly, DreamerV2 performs better on several games when the gradients of the reward predictor are not used to shape the model representations, indicating that representations that are less specific to previously experienced rewards may generalize better.
>
> > What are the current limitations of DreamerV2 in terms of Atari?
>
> We did not observe any systematic bottlenecks of DreamerV2 on Atari compared to the model-free algorithms we compared to. However, some Atari games require directed exploration to learn successful policies, such as Montezuma's Revenge, Private Eye and Pitfall. We also hypothesize that incorporating forms or temporal abstraction and long-term memory are promising directions for future work.
>
> > What lessons can we learn about applying it outside Atari
>
> While we focus on Atari in this paper, preliminary experiments on DeepMind Control continuous control tasks have shown faster learning of DreamerV2 compared to DreamerV1.

---

### Official Review · AnonReviewer4 · 2020-10-28
**Simple and impactful innovation**

**Rating:** 9
**Confidence:** 5

**Review:**

The authors introduce DreamerV2, a modification of the influential Dreamer RL agent (hereafter refered to as DreamerV1). The primary changes from DreamerV1 are a discrete latent space and a modified loss function (and with it, a modified optimization scheme). As in DreamerV1, the agent trains a world model with environment experience, and the policy is learned by "imagining" within the learned latent space using the world model to simulate transitions and rewards. They demonstrate superior performance over a variety of successful benchmarks that use similar compute (1 GPU, 1 environment -- e.g. MuZero, which requires vastly more, is not considered) on Atari. Further, they analyze several ways of aggregating Atari scores, and (while their algorithm performs best of those tried in each aggregation), they recommend one aggregation method (along with several other choices made for benchmarking) going forward.

This is impressive work. The modification over DreamerV1 is simple (simple enough that they can describe important optimization details within the main body of the paper -- great!) and the results are a convincing demonstration of its utility. The methods are detailed and well-described. Further, I think the effort spent on exactly how to benchmark (and in particular, to report scores) is very welcome and useful to communicate (though I'll be interested to hear if others object to portions of the recommendations!).

The main weakness: I wish we gained a better understanding of why the discrete latent space works well. Interpretability might be difficult (and I don't necessarily consider that bad -- the results speak for themselves) but we are left with a mix of several plausible hypotheses. Perhaps some visualization might be useful.

Regardless, I strongly advocate for acceptance. What they propose is relatively simple (and the paper describes it well) and appears to work well in this setting. I'll be recommending students try it in other settings. Further, the benchmarking discussion is very useful for the community.

On the use/interpretation of the discrete latent, a question for the authors: do we have a good sense of what sorts of environments DreamerV2 does poorly on, relative to the continuous latent ablation? DreamerV1 does well on a variety of continuous control tasks -- does DreamerV2 do poorly in continuous control? (Apologies if I missed this described somewhere.) Having some delineation of where it does well vs poorly could help me get a better sense of use of the discrete latent.

Appendix A is most welcome.

----------------------------------

Edit after reading other conversations.

I do think the other reviewers make some fair points. I've adjusted my score to a 9, though, and I'd very much like to see the paper accepted. Here is my thinking on a couple points that led to this score adjustment.

AnonReviewer2's SimPLE-like ablation request: If I'm understanding this correctly, the reviewer would like to see the policy model trained on reconstructions instead of the latent space (or maybe would like the world model trained to future predict in pixel space? not completely clear to me). To me, this would be a potentially insightful ablation, but I think not a dealbreaker that they do not. Two points on this:

(1) If I am understanding the paper and conversations, SimPLE significantly underperforms in the metric (Atari "end performance" under certain normalizations) that the authors care about (and is a fairly established metric). I, then, don't see a *particularly* strong motivation for careful ablations of the method.

(2) Pixel-based future predictions generally perform poorly, and this is I think fairly widely thought to be a strong contributor to the failure of model-based approaches. Again, it would be nice to see that happen here (and it might be insightful to see the quality of the frame predictions) but I think there is a reasonable expectation that this would work poorly.


AnonReviewer1's thoughts on the value of this sort of work in ICLR: I see your point, but I personally think there's great value to this sort of work in this sort of venue. End performance on Atari has been (for better or worse) an important baseline for the field, and the creation of performant model-based approaches has been a central question for several years, now. The proposed improvements over DreamerV1 (which has seen fruitful applications in other work) might be simple, but DreamerV1 did not work well on this baseline, and this does. I think that we far too often get excited by complex new methods, often evaluated with novel and poorly understood baselines and metrics, only to drop them as time goes on. That the innovation is a simple one should make us excited to try it in other applications.

I also think that their discussion of evaluation metrics is very useful -- we continually need more careful conversations about the right ways to measure success. So, in short, the paper might be a technically simple innovation, but it puts forth strong evidence that the method is useful.

---

> ### Author Response · Authors · 2020-11-19
> **Response to AnonReviewer4**
>
> Thank you for your review! We have conducted an ablation study of the world model and included it in the paper. Please find our detailed response below and let us know if you have any further questions or suggestions.
>
> > This is impressive work. The modification over DreamerV1 is simple (simple enough that they can describe important optimization details within the main body of the paper -- great!) and the results are a convincing demonstration of its utility. The methods are detailed and well-described. Further, I think the effort spent on exactly how to benchmark (and in particular, to report scores) is very welcome and useful to communicate (though I'll be interested to hear if others object to portions of the recommendations!).
>
> Thank you!
>
> > On the use/interpretation of the discrete latent, a question for the authors: do we have a good sense of what sorts of environments DreamerV2 does poorly on, relative to the continuous latent ablation?
>
> We completely agree that understanding in which environments the components of our algorithm, and especially the choice of discrete latent variables, are beneficial is interesting. We have updated the paper to include the learning curves for discrete and continuous latents for the 55 individual tasks. We find that discrete latents match or outperform continuous latents on almost all games (the exceptions being Asteroids, Demon Attack, Frostbite, Solaris). This result suggests that the benefit of discrete over continuous latents may not be specific to properties of individual Atari games.
>
> An interesting direction to further investigate the effects of discrete latent variables would be to leverage the offline experience datasets of Agarwal et al. (2020) or Gulcehre et al. (2020) to compare the accuracy of model predictions. This would enable separating the effects of model accuracy from those of improved policy optimization and exploration. We leave such a study for future work.
>
> > DreamerV1 does well on a variety of continuous control tasks -- does DreamerV2 do poorly in continuous control?
>
> While we focus on Atari in this paper, preliminary experiments on DeepMind Control continuous control tasks have shown faster learning of DreamerV2 compared to DreamerV1.

---

### Official Review · AnonReviewer2 · 2020-10-29
**An application of world models to Atari with an unclear motivation**

**Rating:** 4
**Confidence:** 4

**Review:**

The authors build on the Dreamer architecture, that is able to learn models of an environment, to build DreamerV2, which learns a model of an environment in latent space. The authors then train their agent in this latent space. DreamerV2 was evaluated on the Atari learning environment and results showed that it was comparable to Rainbow and better, under certain metrics.

I am unclear on the motivation of this paper. As with previous papers on model-based learning for Atari (i.e. Kaiser et. al (2019)), the goal of learning a model has been to reduce the number of environment steps. However, the authors use the same number of environment steps with the only difference being the model is trained in latent space. Training the model in latent space can speed up learning. Is this the main contribution of the paper?

There is no analysis as to why using a world model for training might lead to better results than training in the real-world if the same number of environment steps are used. What is the authors' perspective on this? Did DreamerV2 use more steps in the world-model environment than in the real-world environment?

Given that the latent space is trained based on some reconstruction error (instead of only being useful to a value function as with value prediction networks) it is not immediately obvious that the latent space will be a better place to learn a policy. Have the authors tried training their method on the reconstructions? Perhaps this will be slower, but I think it is still relevant since the authors claim that it may be easier to learn a model in latent space: "Predicting compact representations instead of images can reduce the effect of accumulating errors"

In the introduction, the authors say: Several attempts at learning accurate world models of Atari games have been made, without achieving competitive performance (Oh et al., 2015; Chiappa et al., 2017; Kaiser et al., 2019)." I do not think this is a fair statement because papers such as Kaiser et al., 2019 intentionally use fewer environment steps.

---After rebuttal---

The authors have partially addressed my concerns, however, I am still not quite sure why their method would be better than SimPLE. The authors' ablation study of removing image gradients does not address my main question about where the performance benefit is coming from. I am assuming that the architecture and hyperparameters that the authors use are different than SimPLE. I think one must instead replace predicting the latent state with the image as SimPLE did to see if that makes a difference in performance. Therefore, I will be keeping my review the same.

---

> ### Author Response · Authors · 2020-11-19
> **Response to AnonReviewer2, Part 2/2**
>
> > Given that the latent space is trained based on some reconstruction error (instead of only being useful to a value function as with value prediction networks) it is not immediately obvious that the latent space will be a better place to learn a policy. [...] Have the authors tried training their method on the reconstructions?
>
> The reconstructions are computed by the image decoder, which only receives the corresponding model state as input. Therefore, all information that is contained in the reconstruction is already contained in the model state. Thus, despite a vast increase in computational requirements, it would be highly unlikely that training the policy from model predictions would be beneficial compared to directly training from the model status.
>
> To further investigate your question, we have conducted an ablation study that learns the model representations only via image prediction or reward prediction, as opposed to using both. We find that DreamerV2 heavily relies on image gradients. Interestingly, removing reward gradients slightly improved performance on several games, suggesting that representations that are less specific to previously received rewards may generalize better.
>
> > Perhaps this will be slower, but I think it is still relevant since the authors claim that it may be easier to learn a model in latent space: "Predicting compact representations instead of images can reduce the effect of accumulating errors"
>
> We agree that the claim "Predicting compact representations instead of images can reduce the effect of accumulating errors" is not studied in detail. We have rephrased this statement in our paper to "Predicting compact representations instead of images has been hypothesized to reduce the effect of accumulating errors".
>
> References:
> - Kaiser et al. Model-Based Reinforcement Learning for Atari. ICLR 2020.
> - Schulman et al. Proximal policy optimization algorithms. 2017.
> - Kostrikov et al. Image Augmentation Is All You Need: Regularizing Deep Reinforcement Learning from Pixels. 2020.

---

> ### Author Response · Authors · 2020-11-19
> **Response to AnonReviewer2, Part 1/2**
>
> Thank you for your review! Previous world models, such as SimPLE (Kaiser et al., 2019), have been far from being competitive with model-free algorithms in terms of final performance on Atari. Our paper is the first to overcome this barrier by introducing an efficient world model that leverages a categorical latent space and KL balancing, achieving a 11x score improvement over SimPLE. We have updated the paper to include an ablation study that highlights the importance of key components of the DreamerV2 world model. We discuss these points in detail below. Please let us know if this fully addresses your concerns, or if there are other issues that we should address.
>
> > In the introduction, the authors say: Several attempts at learning accurate world models of Atari games have been made, without achieving competitive performance (Oh et al., 2015; Chiappa et al., 2017; Kaiser et al., 2019)." I do not think this is a fair statement because papers such as Kaiser et al., 2019 intentionally use fewer environment steps.
>
> Previous approaches based on world models, including SimPLE (Kaiser et al., 2019), have not been able to compete with model-free algorithms in terms of final performance on Atari. The performance of SimPLE only increases up to 500k agent steps, corresponding to 2M environment frames ("results for settings with 20K, 50K, 200K, 500K and 1M [...] results improve until 500K samples").
>
> Over the 26 games SimPLE was evaluated ("selected on the basis of being solvable with existing state-of-the-art model-free deep RL algorithms"), it achieves a gamer normalized median task score of only 0.14 (also see Figure 5 of Kostrikov et al., 2020). Thus, their decision to focus on the low-data regime was not just motivated by the computational requirements of their algorithm but also its low final task performance.
>
> In contrast, we focus on the well-established benchmark of 200M frames, where competitive and tuned model-free methods such as Rainbow and IQN are available for comparison. Our agent is the first that leverages a separately trained world model to achieve human-level Atari performance with a gamer normalized median score of 1.59, corresponding to a 11x improvement in task performance compared to SimPLE.
>
> > I am unclear on the motivation of this paper. As with previous papers on model-based learning for Atari (i.e. Kaiser et. al (2019)), the goal of learning a model has been to reduce the number of environment steps. However, the authors use the same number of environment steps with the only difference being the model is trained in latent space. Training the model in latent space can speed up learning. Is this the main contribution of the paper?
>
> Beside their potential for improving sample-efficiency, improving world models is an impactful research direction due to several practical downstream applications. This includes efficient transfer to previously unseen tasks, directed exploration to reduce model uncertainty, and generalization in offline RL. Please see our introduction section for references of these applications.
>
> These applications have been held back by the difficulty of scaling world models to challenging tasks, such as the Atari benchmark. Our paper is the first to break this barrier and successfully scale world models to human-level performance on Atari, vastly exceeding the performance of previous world models on the benchmark. We achieve this by introducing a world model with categorical latents and KL balancing that offers both accurate and efficient predictions for policy optimization.
>
> > There is no analysis as to why using a world model for training might lead to better results than training in the real-world if the same number of environment steps are used. What is the authors' perspective on this? Did DreamerV2 use more steps in the world-model environment than in the real-world environment?
>
> Following the standard Atari benchmark, our agent sees 200M environment frames. Every 16 environment steps, our agent trains its policy on 2500 imagined model trajectories with a length of 15 steps each. The agent thus learns the policy from 468B imagined model steps, which is about 10,000 higher than the 50M inputs received from the environment after action repeat. We have added this information to the experiments section.
>
> Compared to learning a policy from experience replay, learning the policy from model predictions is not only more computationally efficient because of the compact latent states, it also allows training the policy on imagined states that the model generalizes to and that were not observed in the environment.

---

### Decision · Program_Chairs · 2021-01-07
**Final Decision**

**Decision:**

Accept (Poster)

**Comment:**

The main contribution of the paper is showing that a model-based approach can be competitive with (and even outperform) strong model-free methods on the 200M Atari benchmark. This is achieved through a set of improvements over the original Dreamer algorithm.

Reviewers have been polarized over this submission (4,5,8,9). After reading the paper, reviews, rebuttals, and engaging with all reviewers in private conversations, I am recommending acceptance as a poster. I agree with R3 and R4 that « this is impressive work », « results are a convincing demonstration of its utility », « it is an important setup from the perspective of model-based RL », « the model is elegant », and « the benchmarking discussion is very useful for the community ».

Although it is true, as R1 puts it, that this work can be seen as « an incremental set of tricks over a prior published approach », these tricks are not obvious and lead to very substantial empirical performance gains. Since the authors described them in details and have also committed to sharing their code, I expect them to be quite valuable to other researchers.

Finally, although I respect R2’s choice to stick to their rating of 4, I believe that their main concern, related to not fully understanding why this work improves on the existing SimPLE algorithm, is indeed justified, but is not enough for rejection. DreamerV2 has a lot of differences compared to SimPLE and it would be very costly to investigate in details the impact of each of them. Hopefully, this work will motivate further research in model-based RL that will shed more light on such questions. I would encourage the authors, however, to elaborate a bit more on the differences vs. SimPLE in the « Related work » section (or Appendix, if there is not enough room in the main text).